# EC-DiT: Scaling Diffusion Transformers with Adaptive Expert-Choice Routing

**Haotian Sun**[1,2*†], **Tao Lei**[1], **Bowen Zhang**[1], **Yanghao Li**[1], **Haoshuo Huang**[1],
**Ruoming Pang**[1], **Bo Dai**[2], **Nan Du**[1†]

[1]Apple AI/ML     [2]Georgia Institute of Technology

{tao_lei2, bowen_zhang4, yanghao_li, haoshuoh, dunan}@apple.com,
haotian.sun@gatech.edu, bodai@cc.gatech.edu

## Abstract

Diffusion transformers have been widely adopted for text-to-image synthesis. While scaling these models up to billions of parameters shows promise, the effectiveness of scaling beyond current sizes remains underexplored and challenging. By explicitly exploiting the computational heterogeneity of image generations, we develop a new family of Mixture-of-Experts (MoE) models (EC-DiT) for diffusion transformers with expert-choice routing. EC-DiT learns to adaptively optimize the compute allocated to understand the input texts and generate the respective image patches, enabling heterogeneous computation aligned with varying text-image complexities. This heterogeneity provides an efficient way of scaling EC-DiT up to 97 billion parameters and achieving significant improvements in training convergence, text-to-image alignment, and overall generation quality over dense models and conventional MoE models. Through extensive ablations, we show that EC-DiT demonstrates superior scalability and adaptive compute allocation by recognizing varying textual importance through end-to-end training. Notably, in text-to-image alignment evaluation, our largest models achieve a state-of-the-art GenEval score of 71.68% and still maintain competitive inference speed with intuitive interpretability.

## 1 Introduction

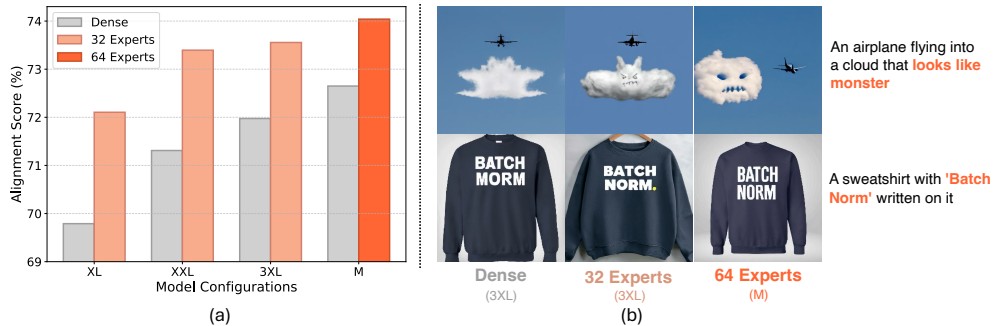

Figure 1: **Performance of EC-DiT.** (a) Across four model configurations, EC-DiT consistently demonstrates superior performance in text-to-image alignment compared to the baseline models with similar activated parameters per prediction. (b) Scaling up with EC-DiT improves text-to-image alignment and visual detail rendering. The alignment score in (a) is the average of GenEval (Ghosh et al., 2023) and DSG scores (Cho et al., 2024).

Diffusion models (Ho et al., 2020; Rombach et al., 2022; Podell et al., 2023) have demonstrated remarkable success in generation across various modalities. There has been growing interest in scaling these models to billions of parameters to improve performance and control computation

---

*Work done while interning at Apple.

†Corresponding authors.

over the generation process. One of the primary focuses of scaling diffusion models is to enhance scalability for *text-conditioned image generation*. As a promising approach for this synthesis task, diffusion transformers (DiT) (Peebles & Xie, 2023) combine the strengths of diffusion models and transformers and exhibit great potential in scalability. Current DiT-based models, such as Stable Diffusion 3 (SD), reach up to 8 billion parameters (Esser et al., 2024) and achieve promising performance in text-to-image generation. However, the effectiveness and optimal approaches for scaling DiTs beyond this size remain underexplored, primarily due to the significant impacts on training and inference efficiency associated with larger models.

The sparse Mixture-of-Experts (MoE) technique (Zoph et al., 2022; Lepikhin et al., 2020; Du et al., 2022; Zhou et al., 2022) has proven to be an efficient method for scaling diffusion models like DiT. Sparse MoEs effectively scale up model capacity while maintaining relatively low computational overhead. In MoE architectures, the router selects a subset of parameters (experts) to process each group of token embeddings and then combines the final outputs from each expert. Current works (Yatharth Gupta, 2024; Fei et al., 2024; Xue et al., 2023) on sparse DiT often adopt a token-choice routing strategy, which is initially proposed for scaling language models (Zoph et al., 2022; Lepikhin et al., 2020; Fedus et al., 2021). Despite technical differences, these methods, in principle, rely on routing *each image token* to the top-ranked experts. However, this routing strategy is suboptimal with diffusion models, which requires a more adaptive routing approach. Specifically, unlike autoregressive generation, DiT processes the entire image sequence at a time. This suggests that the global information of generated samples is accessible throughout the denoising procedure. Additionally, different image areas often contain varying levels of detail. Thus, an optimal router should adaptively allocate heterogeneous computation for different patterns.

Driven by these motivations, we propose scaling DiT with adaptive expert-choice routing (EC-DIT), which is of independent interest, and tailor it for text-to-image generation. This approach leverages global information from each image to optimize computational resource allocation that is aligned with varying image complexities, thus naturally aligning with the diffusion procedure. We scale EC-DIT up to 97 billion parameters with 64 experts. Figure 1 showcases the promising scalability of EC-DIT. Despite a less than 30% increase in computational overhead, our approach significantly improves training convergence, text-to-image alignment, and overall generation quality compared to dense and sparse variants with conventional token-choice MoEs (Yatharth Gupta, 2024; Fei et al., 2024). Notably, in text-to-image alignment evaluation, EC-DIT achieves a new GenEval score of 71.68% while maintaining competitive inference speed compared to other strong baselines. Our main contribution is summarized as follows:

- **Scaling DiT with adaptive computation.** We introduce EC-DIT, a sparsely scaled DiT incorporating expert-choice routing for text-to-image synthesis. This novel approach leverages global context information at each generation step to achieve heterogeneous compute allocation adaptive to different patterns within the generated images.

- **Promising performance at scale.** We examine EC-DIT's scalability and effectiveness by scaling up to 97 billion parameters. At larger model scales, EC-DIT demonstrates faster loss convergence, improves image quality, and reaches state-of-the-art text-image alignment. Moreover, scaling with more experts consistently improves performance across various dimensions without significantly increasing computational overhead.

- **Comprehensive experiments and analysis.** We conduct extensive experiments to validate EC-DIT's superior scalability compared to dense and token-choice variants. Visualizations further demonstrate the model effectively learns an adaptive computation allocation strongly aligned with textual significance.

## 2 METHODOLOGY

Starting with preliminaries of rectified flow model and MoE, this section presents the main design and properties of our EC-DIT models.

### 2.1 PRELIMINARIES

**Image generation via rectified flow.** DiT processes images as sequences of patches in latent space. An input image is first equally divided into a grid of patches, then flattened and projected

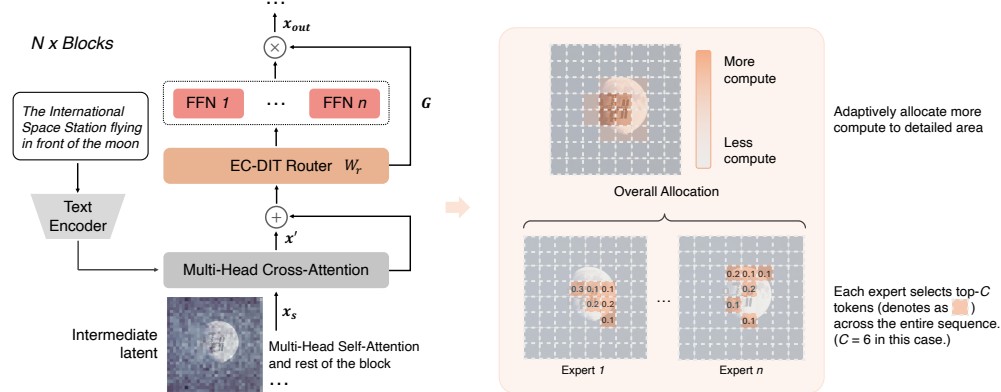

Figure 2: **EC-DIT architecture.** The router leverages information from the entire sequence to adaptively route the most suitable tokens to each expert. Through this heterogeneous routing, more computation is allocated to detailed image areas, such as the `space station` and `moon`, while less computation is used for rendering the background.

into a sequence of embeddings. We denote $\mathbf{x} \in \mathbb{R}^{S \times d_x}$ as the image sequence of length $S$ in latent space, with $d_x$ representing the model's hidden dimension. Diffusion models progressively perform time-dependent denoising procedures on the image sequence $\mathbf{x}$. Given the observation $\mathbf{x}_0 \sim p_0(\mathbf{x})$ from data distribution and noise $\mathbf{x}_1 \sim p_1 = \mathcal{N}(0, I)$, rectified flow (Liu et al., 2022; Lipman et al., 2022; Esser et al., 2024) transfers between two distributions $p_0$ and $p_1$ with a straight-line path $\mathbf{x}_t = t\mathbf{x}_1 + (1 - t)\mathbf{x}_0$. This construction induces a probability flow ordinary differential equation (ODE) $d\mathbf{x}_t = v_\theta(\mathbf{x}_t, t)dt$, where $v_\theta(\mathbf{x}_t, t)$ represents the velocity direction of the path, parameterized by the DiT backbone. Since $v_\theta(\mathbf{x}_t, t) = \frac{d\mathbf{x}_t}{dt} = \mathbf{x}_1 - \mathbf{x}_0$, we optimize the following L2 objective (Liu et al., 2022; Esser et al., 2024):

$$\mathcal{L}(\theta) = \mathbb{E}_{t \sim \pi(t), \mathbf{x}_1 \sim \mathcal{N}(0,I)} \left[ \|(\mathbf{x}_1 - \mathbf{x}_0) - v_\theta(t\mathbf{x}_1 + (1 - t)\mathbf{x}_0, t)\|^2 \right], \quad (1)$$

where $\pi(\cdot)$ refers to the timestep density during training, for which we select the logit-normal sampling strategy (Atchison & Shen, 1980): $t = \log \frac{u}{1-u}$, with $u \sim \mathcal{N}(\mu_t, \sigma_t)$. The hyperparameters $\mu_t$ and $\sigma_t$ control the frequency for intermediate timesteps being sampled during the training process.

**Mixture of Experts.** Shazeer et al. (2017) first introduced the MoE layer that consists of a set $\{\mathcal{E}_i(x)\}_{i=1}^{E}$ of $E$ experts (each of which is often a FeedForward network (FFN)) and a learnable router with the weight $\mathbf{W}_r$. For a given token representation $\mathbf{x}$, the router selects the highest top-$k$ experts based on the gating value of $\mathbf{x} \cdot \mathbf{W}_r$. The output of the MoE layer will then be the weighted combination of the selected expert's computation where the expert weights are the normalized gating values via the softmax distribution. Since the token chooses its best set of experts, this top-$k$ routing is also termed as the token-choice routing.

## 2.2 TEXT CONDITIONING WITH CROSS-ATTENTION MODULES

In the text-to-image synthesis, the model generates image samples $\mathbf{x}$ conditioned on the text prompt $\mathbf{y}$. We utilize a text encoder $\text{Enc}(\mathbf{y}) \in \mathbb{R}^{L \times d_y}$ to extract the text embeddings of $\mathbf{y}$ where $L$ is the number of text tokens, and $d_y$ is the text's hidden dimension. Following the approach in Chen et al. (2023a), we incorporate a multi-head cross-attention layer (Vaswani et al., 2023) between the self-attention and FFN layers in each DiT block. The cross-attention $\mathbf{C}_h$ for head $h$ is written as:

$$\mathbf{Q}_x = \mathbf{x} \cdot \mathbf{W}_x^{\text{query}}, \quad \mathbf{K}_y = \text{Enc}(\mathbf{y}) \cdot \mathbf{W}_y^{\text{key}}, \quad \mathbf{V}_y = \text{Enc}(\mathbf{y}) \cdot \mathbf{W}_y^{\text{value}}, \quad (2)$$

$$\mathbf{C}_h = \text{softmax}\left(\frac{\mathbf{Q}_x \cdot \mathbf{K}_y^\top}{\sqrt{d_x}}\right) \cdot \mathbf{V}_y, \quad (3)$$

where $\mathbf{Q}_x$ is the query matrix for $\mathbf{x}$, and $\mathbf{K}_y, \mathbf{V}_y$ are the key and value matrices for $\mathbf{y}$, respectively; $\mathbf{W}_x^{\text{query}} \in \mathbb{R}^{d_x \times d_x}$ and $\mathbf{W}_y^{\text{key}}, \mathbf{W}_y^{\text{value}} \in \mathbb{R}^{d_y \times d_x}$ are the projection matrices for the image and text representations. Then, the outputs from all $H$ heads are concatenated and linearly transformed by $\mathbf{W}^{\text{out}}$ to obtain the final output, *i.e.*, $\text{MHCA}(\mathbf{x}) = \text{concat}([\mathbf{C}_1, \dots, \mathbf{C}_H]) \cdot \mathbf{W}^{\text{out}}$. This modification effectively injects textual information into the image generation.

## 2.3 SCALING DIT WITH EXPERT-CHOICE ROUTING

Because the majority of computation takes place in the dense FeedForward (FFN) layers (Peebles & Xie, 2023) of DiT, replacing the FFN layer with an MoE layer is one efficient method of scaling. Introducing the MoE layer decouples the effective computation from the model capacity. That is, we can effectively scale up the model capacity without significantly sacrificing the inference speed since only a selected subset of experts will be activated conditioned on the input representation. While this approach seems plausible and effective, it can be further improved for the following reasons:

- **Uniform computation.** The computation assigned to each input token is the same regardless of its representation even though the selected set of experts can be different. However, different image patches naturally demand varying amount of compute, e.g,. foreground needs more attention while background and low-frequency areas may be disregarded.
- **Local context.** Token-choice routing cannot access to the global information of the sequence due to the constraint of auto-regressive decoding. However, this global information can be better exploited in the diffusion process.
- **Load balancing.** To improve the load balance of each expert, some sorts of auxiliary load balance loss (Shazeer et al., 2017; Lepikhin et al., 2020; Du et al., 2022; Zoph et al., 2022) are often required for token-choice routing to work properly.

Therefore, we propose to scale up the diffusion models by better exploiting the structure of the diffusion process. Specifically, following Peebles & Xie (2023); Chen et al. (2023a), we embed the timestep using a two-layer MLP with dimensionality equal to the hidden dimension $d_x$, and feed this embedding into each AdaLN layer. Additionally, as shown in Figure 2, let $\mathbf{x}_s$ be the output of the self-attention module, and $\mathbf{x}'$ be the input to the router, *i.e.*,

$$\mathbf{x}' = \mathbf{x}_s + \texttt{MHCA}(\mathbf{x}_s) \in \mathbb{R}^{S \times d_x}. \tag{4}$$

Therefore, the input sequence $\mathbf{x}\prime$ contains timestep information and integrates different modalities through equation 3.

Each expert is a two-layer FeedForward component, where the $i$-th expert is represented as $\mathcal{E}_i(\mathbf{x}) = \texttt{GeLU}(\mathbf{x} \cdot \mathbf{W}_1^i) \cdot \mathbf{W}_2^i$. Here, $\mathbf{W}_1^i \in \mathbb{R}^{d_x \times d_x'}$ and $\mathbf{W}_2^i \in \mathbb{R}^{d_x' \times d_x}$ are the weight matrices for the $i$-th expert. For each MoE layer, the router is parameterized by the expert embedding $\mathbf{W}_r \in \mathbb{R}^{d_x \times E}$ where $E$ is the total number of experts of the layer. Given the input $\mathbf{x}'$, the router first produces a token-expert affinity score tensor $\mathbf{A} \in \mathbb{R}^{S \times E}$ via $\texttt{softmax}$ along the expert dimension, *i.e.*,

$$\mathbf{A}_{s,i} = \frac{\exp\left((\mathbf{x}' \cdot \mathbf{W}_r)_{s,i}\right)}{\sum_{i=1}^{E} \exp\left((\mathbf{x}' \cdot \mathbf{W}_r)_{s,i}\right)}. \tag{5}$$

This affinity score tensor assesses the relevance between each pair of expert and input token. Unlike the token-choice routing where each token $\mathbf{x}'$ selects the top-$k$ experts from $\mathbf{A}_{s,i}$, EC-DIT works from the expert-choice (Zhou et al., 2022) view where each expert independently selects the top-$C$ tokens in the descending order from $\mathbf{A}_{s,i}$, and $C = S \times f_c/E$ represents the average capacity of each expert, where $f_c$ denotes the capacity factor and reflects the average number of experts assigned to process each token.

Comparing to the token-choice routing, which assigns each token independently, EC-DIT selects the most relevant $C$ tokens from the *entire* sequence. To achieve this, we compute the gating tensor $\mathbf{G} \in \mathbb{R}^{S \times E}$ as follows:

$$\mathbf{G}_{s,i} = \begin{cases} \mathbf{A}_{s,i}, & \mathbf{A}_{s,i} \in \texttt{top-k}\left(\{\mathbf{A}_{s,i} \mid 1 \leqslant s \leqslant S\}, \texttt{k} = C\right) \\ 0, & \text{otherwise}, \end{cases} \tag{6}$$

where $\mathbf{G}_{s,i}$ is the weighting score representing expert $i$'s preference over the $s$-th token.

With equation 3 and equation 6, EC-DIT directs the computational focus of the experts towards tokens with significant textual information and visual patterns, while also being adaptive to varying denoising timesteps. Specifically, such allocation is not one-to-one. For instance, as shown in Figure 2, tokens corresponding to the $\texttt{moon}$ and $\texttt{space station}$ contain substantial detail and are therefore assigned to multiple experts for finer processing. In contrast, tokens representing a plain background may not be assigned to any experts at all. This makes the computation more adaptive and efficient, concentrating resources where they are most needed.

---

**Algorithm 1** Pseudocode of EC-DIT's Routing Layer

---

```
# B: batch size, S: sequence length, d: hidden dimension
# E: number of experts, C: expert capacity
# experts: list of length E containing expert FFNs
def ec_dit_routing(x_p, W_r, experts):
    # 1. Compute token-expert affinity scores
    logits = einsum('bsd,de->bse', x_p, W_r)        # shape: (B, S, E)
    affinity = softmax(logits, dim=-1)              # shape: (B, S, E)
    affinity = einsum('bse->bes', affinity)         # shape: (B, E, S)
    # 2. Select the top-k tokens for each expert
    gating, index = top_k(affinity, k=C, dim=-1)    # shape: (B, E, C)
    dispatch = one_hot(index, num_classes=S)        # shape: (B, E, C, S)
    # 3. Process the tokens by each expert and combine
    x_in = einsum('becs,bsd->becd', dispatch, x_p)  # shape: (B, E, C, d)
    x_e = [experts[e](x_in[:, e]) for e in range(E)]
    x_e = stack(x_e, dim=1)                         # shape: (B, E, C, d)
    x_out = einsum('becs,bec,becd->bsd', dispatch, gating, x_e)
    return x_out                                    # shape: (B, S, d)
```

---

We define a set of indexing vector $\{\mathcal{I}_i \mid 1 \leqslant i \leqslant E\}$ to filter the input tokens allocated to each expert, *i.e.*,

$$\mathcal{I}_i = \{s \mid \mathbf{G}_{s,i} > 0, 1 \leqslant s \leqslant S\} \tag{7}$$

The output $\mathbf{x}_{\text{out}} \in \mathbb{R}^{S \times d_x}$ of the sparse layer is then obtained by combining the results from each expert using the gating tensor $\mathbf{G}$, *i.e.*,

$$\mathbf{x}_{\text{out}} = \sum_{i=1}^{E} \left(\mathbf{G}_{\mathcal{I}_i,i}\right)^{\top} \mathcal{E}_i \left(\mathbf{x}'_{\mathcal{I}_i,:}\right). \tag{8}$$

Note that equation 6 inherently ensures full utilization of each expert and perfect load balance by design. Therefore, EC-DIT is trained directly via optimizing equation 1, without the need for an additional load-balancing loss (Zoph et al., 2022). The pseudocode of EC-DIT's routing strategy is presented in Algorithm 1.

**Remark:** The proposed EC-DIT shares several key benefits, making it particularly well-suited for DiT-based diffusion models in text-to-image generation tasks.

- **Adaptive computation.** EC-DIT is implicitly aware of the textual context and denoising stages. As we will demonstrate in Section 3, EC-DIT effectively enables context-aware routing and dynamically adjusts its computation across both model depth and denoising timesteps.
- **Global context information.** Diffusion models, unlike autoregressive models, process the entire sequence at each denoising step. As described in equation 6, EC-DIT leverages this global context, combining textual and image information to optimize token-to-expert allocation.
- **Load balancing.** While token-choice routing often requires auxiliary load-balancing loss, EC-DIT 's routing mechanism inherently ensures balanced utilization of all experts. This prevents bottlenecks and enhances overall efficiency by allocating computations more effectively.

## 3 EVALUATIONS

In this section, we empirically evaluate EC-DIT's performance across various metrics for text-image alignment and image quality and validate the training and inference efficiency. We then visualize the heterogeneous compute allocation within the proposed EC-DIT. Finally, we present comparative results against dense and token-choice baselines. Throughout the paper, we use the naming convention DENSE-<Config> for dense models and EC-DIT-<Config>-<n>E, where <Config> is the model configuration name; <n> is the number of experts.

### 3.1 EXPERIMENT SETUP

**Model architecture.** Following Chen et al. (2023a), we adopt a modified DiT architecture with additional cross-attention modules for text-to-image generation. For all models evaluated, we use a 670M clip-based text encoder with the T5 tokenizer (Raffel et al., 2023) for the text input and a 34M variational autoencoder (VAE) with 8 channels (CLIP-ViT-bigG) for the image input (Hessel et al., 2022). Both encoders remain frozen during DiT training. The transformer component is configured with four model sizes: XL, XXL, 3XL, and M, as detailed in Table 1. Starting from

Table 1: **Model size and specifications** across four backbone configurations. For **Total Params.**, $n$E denotes EC-DɪT with $n$ experts in a single MoE layer. **Activated Params.** refer to the average number of activated parameters per token in the forward pass. The detailed calculation is depicted in Appendix D. In **Model Arch.**, #Head is the number of query heads, and #KV represents the number of key/value heads for grouped-query attention (GQA). [*]Note that a trade-off was made in the model depth for the M configuration, where DᴇɴsE-M has 46 layers, while EC-DɪT-M has 38 layers in order to be able to fit into the HBM of training accelerators.

| Config. | Total Params. | | | | | Activated Params. | Model Arch. | | | |
|---|---|---|---|---|---|---|---|---|---|---|
| | DᴇɴsE | 8E | 16E | 32E | 64E | EC-DɪT | #Layers | Hidden dim. | #Head | #KV |
| XL | 1.47B | 2.51B | 3.70B | 6.08B | – | 1.62B | 28 | 1,152 | 18 | 6 |
| XXL | 2.35B | 4.87B | 7.73B | 13.47B | – | 2.71B | 38 | 1,536 | 24 | 6 |
| 3XL | 4.50B | 10.74B | 17.87B | 32.15B | – | 5.18B | 42 | 2,304 | 36 | 6 |
| M | 8.03B | – | – | – | 97.21B | 8.27B | 38/46[*] | 3,072 | 48 | 12 |

the second layer, we scale each dense model by interleavingly replacing all even-numbered dense layers with sparse MoE layers, where each expert is an FFN with dimensions identical to the dense counterpart. We set the capacity factor $f_c = 2.0$ throughout the training and inference stages for all sparse models. We scale the XL-3XL dense models with 8, 16, and 32 experts. To further test the model's scalability, we scale the M configuration with 64 experts. To explore the trade-off between model depth and width during scaling, we reduce the number of layers in EC-DɪT-M-64E to 38, while maintaining 46 layers in the DᴇɴsE. With these settings, our EC-DɪT-3XL-32E reaches 32.15 billion parameters, and the largest EC-DɪT-M-64E attains 97.21 billion parameters beyond all the current sparse diffusion models (Fei et al., 2024; Lin et al., 2024; Balaji et al., 2022).

**Training details.** We collect and utilize approximately 1.2 billion text-image pairs from the Internet (McKinzie et al., 2024; Lai et al., 2023). The model resolution is set to $256 \times 256$, with a patch size of $2 \times 2$. To accelerate training, we employ the masking technique proposed in (Zheng et al., 2023) with a masking ratio of 0.5. This results in the input sequence length of 128 per image. Model training is conducted on v4 and v5p TPUs with a batch size 4096. We use the RMSProp with momentum optimizer (Hinton, 2012) with a learning rate of 1e-4 and 20K warmup steps. All models are trained with Distributed Data Parallelism (DDP) or Fully Sharded Data Parallel (FSDP) for 800K steps. The expert dimension of EC-DɪT is fully shared across the TPU mesh. To enhance training efficiency, we regroup the first two dimensions of the input sequence (batch_size, sequence_length) into (outer_batch, num_group, group_size). We maintain a group_size of 1024 for training and 512 for inference, which allows EC-DɪT to process two images in a sequence at inference time.

**Evaluation metrics.** To assess the model's text-to-image capabilities, we primarily use GenEval (Ghosh et al., 2023) and Davidsonian Scene Graph (DSG) scores (Cho et al., 2024). GenEval segments and detects objects in the generated samples to evaluate their alignment with the conditioned texts. For DSG, we use an internal version of Gemini to perform Visual Question Answering (VQA) to assess the text-image alignment. To evaluate the image quality of the generated images, we measure zero-shot Fréchet Inception Distance (FID) (Heusel et al., 2017) along with CLIP Score (Hessel et al., 2022) on the MS-COCO $256 \times 256$ dataset using 30K samples (Lin et al., 2015). We also provide generated samples from a subset of Partiprompts (Yu et al., 2022) in Appendix E.

## 3.2 Tᴇxᴛ-ᴛo-ɪᴍᴀɢᴇ Aʟɪɢɴᴍᴇɴᴛ

We present the comparative evaluation results on GenEval and DSG, both of which are commonly used to measure text-to-image alignment. Table 2 shows the comprehensive GenEval results of EC-DɪT compared to current state-of-the-art models. Specifically, our largest sparse model, EC-DɪT-M-64E (resolution $256 \times 256$), reaches GenEval of 71.68% and surpasses SD3 (resolution $512 \times 512$) in the overall score (68.00%) and most sub-tasks. Notably, the solely pretrained EC-DɪT-M-64E without fine-tuning stage even outperforms the SD3 model fine-tuned with DPO. Notably, both DPO fine-tuning and training with higher resolution positively impact the GenEval score, as

Table 2: **GenEval comparison.** Our largest scaled model (EC-DɪT-M with 64 experts) surpasses all existing models in overall performance and most sub-tasks of the GenEval Metrics (Ghosh et al., 2023), including the DPO-finetuned version of the current state-of-the-art, SD3-Large (Esser et al., 2024) (depth=38, resolution $512 \times 512$). The scores for models preceding ours are sourced from Esser et al. (2024). The **best** and second-best entries are highlighted.

| Model (↓) / Score (%) (→) | Overall | Single obj. | Two obj. | Counting | Colors | Position | Color attr. |
|---|---|---|---|---|---|---|---|
| SD v1.5 (Rombach et al., 2022) | 43.00 | 97.00 | 38.00 | 35.00 | 76.00 | 4.00 | 6.00 |
| PixArt-$\alpha$ (Chen et al., 2023a) | 48.00 | 98.00 | 50.00 | 44.00 | 80.00 | 8.00 | 7.00 |
| SD v2.1 (Rombach et al., 2022) | 50.00 | 98.00 | 51.00 | 44.00 | 85.00 | 7.00 | 17.00 |
| DALL-E 2 (Ramesh et al., 2022) | 52.00 | 94.00 | 66.00 | 49.00 | 77.00 | 10.00 | 19.00 |
| SDXL (Podell et al., 2023) | 55.00 | 98.00 | 74.00 | 39.00 | 85.00 | 15.00 | 23.00 |
| SDXL Turbo (Podell et al., 2023) | 55.00 | **100.00** | 72.00 | 49.00 | 80.00 | 10.00 | 18.00 |
| IF-XL (Saharia et al., 2022) | 61.00 | 97.00 | 74.00 | 66.00 | 81.00 | 13.00 | 35.00 |
| DALL-E 3 (Shi et al., 2020) | 67.00 | 96.00 | 87.00 | 47.00 | 83.00 | **43.00** | 45.00 |
| SD3-Large (Esser et al., 2024) | 68.00 | 98.00 | 84.00 | 66.00 | 74.00 | 40.00 | 43.00 |
| SD3-Large (Esser et al., 2024) w/ DPO | 71.00 | 98.00 | **89.00** | 73.00 | 83.00 | 34.00 | 47.00 |
| Dᴇɴsᴇ-3XL | 68.92 | 99.69 | 86.00 | 69.41 | 81.67 | 20.58 | 56.19 |
| EC-DɪT-3XL-32E | 70.91 | 99.64 | 87.88 | 72.53 | 83.84 | 21.19 | 60.40 |
| EC-DɪT-M-64E | **71.68** | 99.84 | 88.67 | **73.69** | **85.77** | 21.33 | **60.80** |

SD3 generally yields higher GenEval scores equipped with these techniques (Esser et al., 2024). This further underscores the potential benefits of scaling dense model with EC-DɪT. Furthermore, EC-DɪT-3XL-32E, with 5.18B activated parameters, achieves a nearly equivalent GenEval score of 70.91% compared to the 71.00% score from SD3-Large with DPO, with only 64% activation size of the latter. The EC-DɪT-3XL-32E's inference time is only 33.8% of the 8B Dᴇɴsᴇ-M. These results demonstrate that our scaling approach can achieve performance comparable to significantly larger dense models, even without additional fine-tuning.

We further validate the effectiveness of EC-DɪT through DSG scores (Cho et al., 2024), as depicted in Figure 3. EC-DɪT consistently improves DSG performance over the dense variants across different model configurations. Additionally, for each model size, increasing the number of experts further enhances text-image alignment. In both GenEval and DSG evaluations, increasing the model's capacity with the EC-DɪT directly enhances its text-following ability and leads to more accurately aligned image generation. A detailed DSG score breakdown is provided in Appendix B.

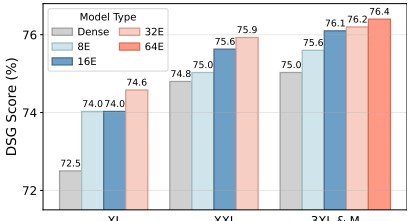

Figure 3: **DSG comparison.** Scaling up with EC-DɪT elevates performance.

## 3.3 INFERENCE-TIME EFFICIENCY

Figure 4 illustrates the inference efficiency of EC-DɪT compared to dense models. Across all base model configurations, EC-DɪT consistently delivers improved performance in text-image alignment. The actual inference overhead introduced by EC-DɪT ranges from 20% to 28% across different configurations. This practical overhead is greater than the theoretical one of up to 15%, which is calculated in Table 1. For EC-DɪT-M, although the theoretical overhead is around 3%, the actual overhead is measured at 23%. This difference might be attributed to the varying efficiency in inference-time parallelism: EC-DɪT-M uses model parallelism to fit on $8 \times$H100 GPUs, whereas the dense model utilizes FSDP. Despite these factors, scaling with EC-DɪT reliably enhances text-to-image alignment up to the state-of-the-art level with less than 30% additional overhead.

## 3.4 HETEROGENEOUS COMPUTE ALLOCATION

In Figure 6, we visualize the heterogeneity of compute allocation through heatmaps that show the number of experts assigned to each image token. We observe that EC-DɪT allocates more computation to areas with clear textual significance, such as major objects or detailed patterns. For example, in Figure 6(a), the main object (such as the moon) and the rendered text receive the most compu-

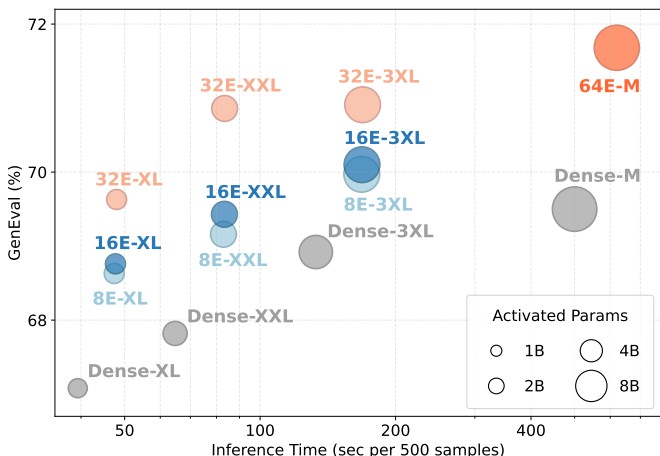

Figure 4: **Inference-time efficiency.** The circle size is proportional to the total activated parameters. Inference time represents the time elapsed to generate 500 samples on $8\times$H100 GPUs. EC-DIT shows superior performance compared to dense models, with less than 30% additional overhead.

Figure 5: **Comparison with token-choice baselines** on FID and CLIP Score of EC-DIT-XXL (`EC`) and GShard (`GS`) with top-2 token-choice routing.

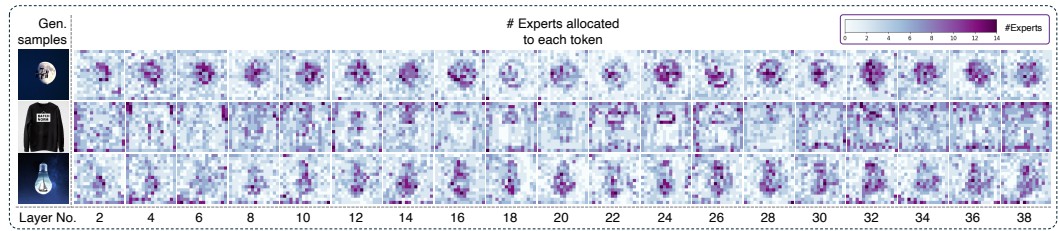

(a) Allocation per sparse layer at the denoising step $t = 40$

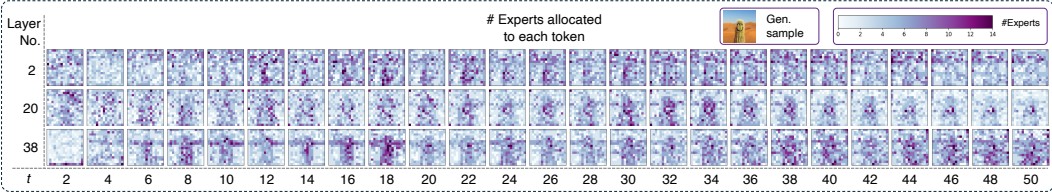

(b) Allocation per denosing timestep $t$

Figure 6: **Visualization of compute allocation** for EC-DIT-XXL-32E (38 layers, with even layers being sparse). Each point in the heatmap represents the number of experts selecting the corresponding token in the latent space. Darker colors indicate areas where more compute is allocated. The heatmaps reveal that the router tends to assign more computations to significant objects, such as the moon in the first sample, and to detailed areas, such as text in the second sample.

tation. In contrast, the background (composed of nearly monotone colors) receives much lighter computation. This allocation heterogeneity tends to be more pronounced in later layers. Figure 6(b) also demonstrates that later layers exhibit heterogeneity in fewer denoising steps. We hypothesize that the early layers handle most "low-frequency" areas, such as the background tokens, and provide context for the later stages. Meanwhile, the routers from later layers more quickly converge to the most textually representative tokens. This coincides with previous observations in pixel space (Lei et al., 2023). Overall, EC-DIT 's compute allocation exhibits high heterogeneity, where a single image token can be processed by up to 44% of the total computation to capture intricate details, while only a few experts process non-detailed areas or even skipped in certain layers. Notably, EC-DIT learns to achieve this adaptive and heterogeneous compute allocation through end-to-end training. We hypothesize that the input to the router, derived from the cross-attention module, likely contains essential cross-modal information from both text and image, which can be effectively leveraged to achieve this adaptive routing.

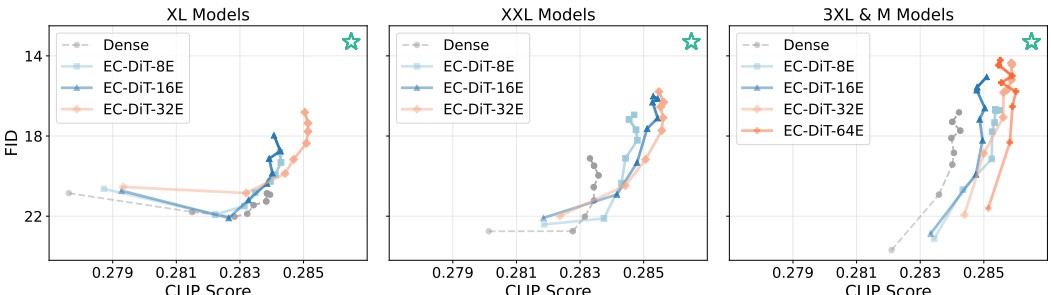

Figure 7: **FID vs. CLIP Score curve during training**. The closer a point is to the upper right corner (☆), the better the model performs in terms of higher image quality (lower FID) and text-image alignment (higher CLIP Score). For each model, eight points are plotted from left to right, representing the model's performance at 100k training step intervals.

### 3.5 COMPARISON WITH TOKEN-CHOICE ROUTING

Figure 5 compares EC-DIT with models scaled using token-choice MoE (Yatharth Gupta, 2024; Xue et al., 2023; Fei et al., 2024). For a fair comparison, we maintained identical experimental settings, except for replacing the routing strategy with token-choice and incorporating an auxiliary load-balancing loss in the training objective. The token-choice baseline uses top-2 routing, which matches the activation size of EC-DIT with a capacity factor of $f_c = 2.0$. Our method consistently demonstrates superior training convergence and performance throughout the entire training period. Notably, EC-DIT with 8 experts rivals the token-choice baseline with 16 experts in both generation quality and text-image alignment, while EC-DIT with more experts significantly outperforms this token-choice baseline. One reason for the slower training convergence in token-choice routing is the equal compute allocation to different image areas, which makes the compute less adaptive. Detailed image areas may not receive sufficient computation to render complex patterns, while simpler areas waste the computation they receive. Moreover, token-choice routing processes each token independently without considering sequence information, leading to a lack of perception of the overall relationships between local tokens. Additionally, the inherent load imbalance in token-choice routing results in some experts being overloaded with an excessive number of tokens and creating a training bottleneck (Zhou et al., 2022). Appendix C provides a comprehensive comparison of EC-DIT and the token-choice baseline regarding GenEval performance. As shown in Table 5, EC-DIT outperforms the token-choice model across all tested configurations, which further validates the advantages of EC-DIT in text-image alignment.

### 3.6 SCALING EC-DIT MODELS WITH MORE EXPERTS

From previous results, scaling a dense model with more experts consistently brings performance gains in both convergence and generation quality. In Figure 8, increasing the number of experts leads to slightly improved loss convergence. Figure 3 and Figure 4 collectively demonstrate that adding more experts to the same dense model consistently enhances the model's text-image alignment capability. Additionally, Figure 4 indicates that increasing the number of experts effectively results in a more powerful model without increasing the activation size nor the inference time, as the expert capacity $C$ is dynamically adjusted based on the number of experts. In Figure 7, we further examine the trend of FID vs. CLIP Score during training. Despite fluctuations in the curve, the general trend shows that adding more experts shifts the curve towards improved image generation quality and text-image alignment. Therefore, we suggest increasing the number of experts if the computational budget allows, as this consistently yields performance gains without additional inference overhead.

### 3.7 TRAINING DYNAMICS.

We also analyze the training dynamics and convergence of dense models and EC-DIT with varying model configurations and numbers of experts, as shown in Figure 8. Across all model settings, we observe a significant improvement in loss reduction between the dense models and sparse EC-DIT throughout the training period. EC-DIT demonstrates a more effective ability to learn denoising for rectified flow generation and leads to better loss convergence. For example, EC-DIT-XXL models trained with 200K steps achieve a training loss comparable to DENSE-XXL models that require

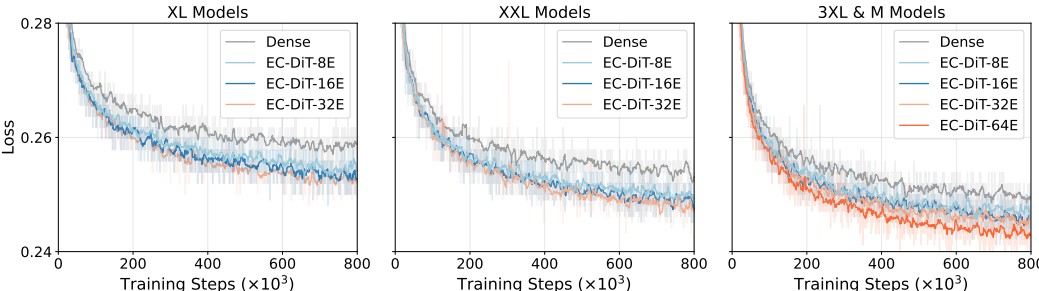

Figure 8: **Training loss of the EC-DⅈT**. The results show a noticeable loss gap between the dense model and the sparse EC-DⅈT. The training loss and convergence speed is further improved by increasing the number of experts.

around 400K steps. Additionally, the sparse EC-DⅈT-XXL models eventually stabilize at a lower loss than their dense counterparts.

## 4 RELATED WORK

**Mixture-of-Experts.** Many MoE models utilize a token-choice routing strategy, where each token independently selects the top-$k$ experts for processing. Sparsely Gated MoE (Shazeer et al., 2017) first introduced top-$k$ gating in LSTM models. Such routing strategy was later extended to transformers for language modeling in GShard (Lepikhin et al., 2020), Switch Transformer (Fedus et al., 2021), GLaM (Du et al., 2022), DeepSpeed-MoE (deepspeed), and ST-MoE (Zoph et al., 2022). To address potential load imbalances among experts, these token-choice approaches typically incorporate an auxiliary load-balancing loss (Shazeer et al., 2017). Alternatively, Zhou et al. (2022) proposed an expert-choice routing in text domain that selects the most suitable text tokens for each expert. Inspired by this, we adapt and scale up the DiT model using expert-choice routing tailored specifically for diffusion-based generative tasks. With this approach, each image token from different image areas receives heterogeneous computation allocation. Additionally, the method inherently achieves perfect load balance, as each expert processes an equal number of image tokens, which optimizes both efficiency and performance for the diffusion process.

**Scaling diffusion models with MoE.** Recent advancements have leveraged MoEs to scale up dense diffusion models for image generation tasks. Several studie have focused on *class-conditioned image generation*. DTR (Park et al., 2023) and Switch-DiT (Park et al., 2024) approach the routing problem as multitask learning and utilize token-choice MoE for different denoising stages. Similarly, MEME (Lee et al., 2023) employs an ensemble of denoising experts specialized in distinct timestep ranges. Furthermore, some models have extended this approach for *text-to-image generation*. ERNIE-ViLG 2.0 (Feng et al., 2022) and eDiff-I (Balaji et al., 2022) employ ensembles of specialized experts for different denoising stages. Using U-Net as their backbone, these methods scale up to 24 billion parameters. However, these models activate only one expert at each timestep, which reduces efficiency at large scales. Recently, some works have shifted towards sparse MoEs. SegMoE (Puigcerver et al., 2023) employs top-k routing to scale up SDXL (Podell et al., 2023) with up to four experts. RAPHAEL (Xue et al., 2023) combines the concepts of distinct timestep experts and token-choice MoEs. DiT-MoE (Fei et al., 2024) applies token-choice routing to produce a sparse DiT with up to 16 billion parameters. Nevertheless, these methods often suffer from load imbalance issues inherent to the token-choice scheme, requiring an additional auxiliary balancing loss to mitigate this problem. The most similar work to ours is MoMa (Lin et al., 2024), which also utilizes expert-choice routing. However, MoMa explores model scaling only on a relatively small scale, with its largest model reaching 7 billion parameters. In contrast, we leveraged the proposed adaptive routing to scale EC-DⅈT up to *97 billion parameters* and demonstrated the DiT scaling potential on a much larger scope.

## 5 CONCLUSION

We propose EC-DⅈT, a novel scaling approach for DiT using expert-choice routing. EC-DⅈT leverages global image information and allocates computation to each image patch adaptive to its pattern complexity. Our model effectively scales up to 97 billion parameters and significantly improves training efficiency, text-to-image alignment, and overall generation quality compared to dense models and sparse DiTs with token-choice routing.

## 6 ETHICS STATEMENT

In this paper, we propose a new scaling strategy for text-to-image generation, improving both the quality of generated images and their alignment with given textual inputs. However, this general approach potentially carries ethical concerns. For example, the generative model could be misused to generate harmful or malicious content, such as imagery based on violent or discriminatory text prompts. We acknowledge these concerns and are committed to addressing them by developing more advanced dataset filtering mechanisms and model calibration techniques. Our future work will prioritize increasing the safety and ethical use of text-to-image generation models.

## 7 REPRODUCIBILITY STATEMENT

In Section 2, we present our method with detailed formulations, examples, and visual demonstrations to clarify the model structure and mechanism (such as Figure 2). In Section 3, we introduce the training dataset and describe the most effective model parameters and components. We also provide specific details regarding the hardware used for training and inference like Figure 4. Additionally, we report the training dynamics of the model, depicted in Figure 8. Through various tables and figures, we present the performance of the proposed method from multiple perspectives, such as Table 2, Figure 3, and Figure 7. We compared our approach with existing methods targeting similar domains or overlapping methodologies, including both theoretical analysis and experimental evaluations (e.g., Figure 5). Notably, all evaluation experiments were conducted on public datasets or benchmarks, such as MSCOCO, GenEval, FID, and CLIP Score. To further ensure reproducibility, we plan to release the model weights contingent on the acceptance of this work.

ACKNOWLEDGMENTS

We would like to thank John Peebles, Xianzhi Du, Chen Chen, Wenze Hu, Riley Roberts, Wei Liu, Zhengfeng Lai, Vasileios Saveris, Peter Grasch, and Yinfei Yang for constructive feedback and discussions.

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

## A FUTURE WORK

In this work, our approach primarily focuses on adaptive computation based on image information, such as pattern complexity. There may also be additional factors, such as object semantics or compositional relationships, that could further improve generation quality, and we plan to investigate how these additional sources of information can be leveraged to further boost EC-DIT's performance.

Additionally, the adaptive routing mechanism of EC-DIT can seamlessly integrate into any sequence- or chunk-wise encoding and generation process. For instance, adaptive routing can be directly applied to the tokenization (Gafni et al., 2022) commonly used in mixed-modal, early-fusion language models (Team, 2024). Similarly, in multi-modal models unifying autoregressive and diffusion paradigms (Zhou et al., 2024), expert-choice routing can be naturally integrated into the diffusion-based image generation component. Furthermore, EC-DIT could potentially enhance long-video generation methods (Chen et al., 2023b; Guo et al., 2023), where the global context within each video chunk could be effectively utilized. We will also explore these directions in our future work.

## B DETAILED DSG SCORES

Table 3: **DSG Score breakdown.** An internal version of Gemini was used to assess each sample's quality through VQA, as proposed in Cho et al. (2024).

| Model (↓) / Score (→) | Overall | Counting | Real user | Text | Paragraph | Poses | TIFA-160 | Relation | Defying |
|---|---|---|---|---|---|---|---|---|---|
| DENSE-XL | 72.50 | 71.50 | 50.00 | 60.70 | 86.70 | 74.80 | 83.70 | 80.40 | 73.90 |
| EC-DIT-XL-8E | 74.03 | 72.63 | 52.13 | 65.90 | 87.10 | 75.63 | 85.83 | 78.70 | 76.20 |
| EC-DIT-XL-16E | 74.03 | 74.20 | 52.17 | 64.57 | 87.00 | 73.50 | 86.13 | 79.60 | 76.63 |
| EC-DIT-XL32E | 74.58 | 73.53 | 52.20 | 66.33 | 86.58 | 74.60 | 86.85 | 80.48 | 79.15 |
| DENSE-XXL | 74.80 | 74.40 | 51.60 | 66.20 | 87.30 | 76.10 | 86.30 | 82.10 | 77.80 |
| EC-DIT-XXL-8E | 75.03 | 74.17 | 53.55 | 67.95 | 86.47 | 73.58 | 87.15 | 81.35 | 78.60 |
| EC-DIT-XXL-16E | 75.63 | 76.33 | 53.60 | 68.52 | 87.40 | 72.80 | 88.18 | 82.40 | 78.60 |
| EC-DIT-XXL-32E | 75.93 | 76.55 | 53.63 | 71.57 | 88.18 | 73.63 | 87.80 | 80.85 | 78.08 |
| DENSE-3XL | 75.03 | 74.47 | 52.83 | 70.33 | 87.73 | 69.40 | 86.77 | 83.00 | 78.10 |
| EC-DIT-3XL-8E | 75.60 | 76.43 | 52.23 | 72.60 | 87.80 | 72.00 | 88.38 | 82.13 | 76.55 |
| EC-DIT-3XL-16E | 76.10 | 75.63 | 54.10 | 73.10 | 88.07 | 73.70 | 88.00 | 81.57 | 77.90 |
| EC-DIT-3XL-32E | 76.20 | 78.27 | 54.47 | 71.33 | 87.63 | 73.17 | 87.90 | 82.00 | 78.13 |
| EC-DIT-M-64E | 76.40 | 77.33 | 54.67 | 72.50 | 88.23 | 74.33 | 87.27 | 82.40 | 77.97 |

Table 4: **DSG comparison with token-choice baselines.** GShard (`GS`) are implemented with two base model configurations (`XL` and `XXL`). GShard uses top-2 token-choice routing.

| Model (↓) / Score (→) | Overall | Counting | Real user | Text | Paragraph | Poses | TIFA-160 | Relation | Defying |
|---|---|---|---|---|---|---|---|---|---|
| GS-XL-8E | 73.47 | 73.27 | 51.43 | 63.73 | 86.00 | 73.50 | 86.70 | 78.67 | 76.37 |
| GS-XL-16E | 73.90 | 74.30 | 52.80 | 62.80 | 86.90 | 74.00 | 85.90 | 81.00 | 74.30 |
| GS-XL-32E | 74.07 | 74.13 | 52.17 | 64.53 | 86.50 | 75.23 | 86.60 | 78.47 | 77.23 |
| GS-XXL-8E | 75.10 | 76.10 | 53.80 | 68.60 | 87.40 | 72.70 | 87.70 | 80.50 | 76.00 |
| GS-XXL-16E | 75.30 | 74.50 | 52.83 | 66.57 | 87.27 | 77.67 | 87.73 | 80.63 | 78.33 |
| GS-XXL-32E | 75.40 | 76.40 | 52.70 | 67.20 | 87.10 | 78.90 | 88.40 | 79.80 | 76.30 |

Table 3 presents the detailed DSG scores for DENSE and EC-DIT across various model configurations. Scaling with EC-DIT consistently improves performance in several categories, such as `Counting` and `Text`, and leads to better overall performance. Table 4 shows that EC-DIT demonstrates superior performance over the token-choice baselines across most evaluation DSG categories.

# C    GENEVAL COMPARISON WITH TOKEN-CHOICE BASELINE

Table 5: **GenEval comparison with token-choice baselines.** EC-DIT (`EC`) and GShard (`GS`) are implemented with two base model configurations (`XL` and `XXL`). GShard uses top-2 token-choice routing.

| Model (↓) / Score (%) (→) | Overall | Single obj. | Two obj. | Counting | Colors | Position | Color attr. |
|---|---|---|---|---|---|---|---|
| DENSE-XL | 67.08 | 99.80 | 82.47 | 64.28 | 82.46 | 19.61 | 53.86 |
| GS-XL-8E | 67.82 | 99.69 | 81.19 | 68.85 | 80.95 | 20.83 | 55.44 |
| GS-XL-16E | 68.00 | 99.71 | 81.64 | 66.39 | 82.85 | 20.56 | 56.84 |
| GS-XL-32E | 68.52 | 99.92 | 82.88 | 69.18 | 83.23 | 20.30 | 55.63 |
| EC-DIT-XL-8E | 68.62 | 99.69 | 83.71 | 68.44 | 81.85 | 19.78 | 58.28 |
| EC-DIT–XL-16E | 68.76 | 99.69 | 83.21 | 66.45 | 82.36 | 22.05 | 58.83 |
| EC-DIT–XL-32E | 69.38 | 99.79 | 84.14 | 68.61 | 83.68 | 21.02 | 59.03 |
| DENSE-XXL | 67.82 | 99.67 | 82.28 | 69.77 | 81.25 | 19.09 | 54.88 |
| GS-XXL-8E | 68.30 | 99.61 | 83.96 | 68.83 | 80.19 | 19.81 | 57.38 |
| GS-XXL-16E | 68.49 | 99.59 | 84.88 | 69.04 | 80.37 | 19.97 | 57.09 |
| GS-XXL-32E | 69.96 | 99.38 | 86.17 | 70.00 | 85.51 | 20.44 | 58.25 |
| EC-DIT–XXL-8E | 69.11 | 99.84 | 85.43 | 68.59 | 81.85 | 20.31 | 58.61 |
| EC-DIT–XXL-16E | 69.43 | 99.34 | 83.46 | 71.62 | 83.83 | 20.94 | 57.42 |
| EC-DIT–XXL-32E | 70.86 | 99.53 | 86.47 | 72.34 | 84.39 | 21.00 | 61.39 |

Table 5 further compares the GenEval performance of EC-DIT with the token-choice baselines (Lepikhin et al., 2020). By leveraging global image information and adaptive computation, EC-DIT demonstrates superior performance over the token-choice baseline across most evaluation categories.

# D    CALCULATING ACTIVATED PARAMETERS INCREMENT

We present the pseudocode of calculating the activated parameters increment in Algorithm 2. Here, the parameters `hidden_dim`, `num_sparse_layers`, `num_heads`, and `num_kv_heads` for each model configuration are specified in Table 1. We set `ffn_factor=4.0` and `capacity_factor=2.0` globally.

---

**Algorithm 2** Pseudocode for Activated Parameters Increment Calculation

---

```
# Dense FFN Size
dense_ffn_size = 2 * hidden_dim * hidden_dim * ffn_factor

# Router Size
router_size = num_experts * hidden_dim

# Attention Size
attn_size = hidden_dim * (num_heads + num_kv_heads * 2 + num_heads) * attn_key_dim

# Activated Dense Total
activated_dense_total = (attn_size + dense_ffn_size) * num_sparse_layers

# Activated Sparse Total
activated_sparse_total = (
    attn_size
    + 2 * hidden_dim * hidden_dim * ffn_factor * capacity_factor
    + router_size
) * num_sparse_layers

# Activated Total Increment
activated_increment = activated_sparse_total - activated_dense_total
```

---

# E   MORE GENERATED SAMPLES AND VISUAL COMPARISONS

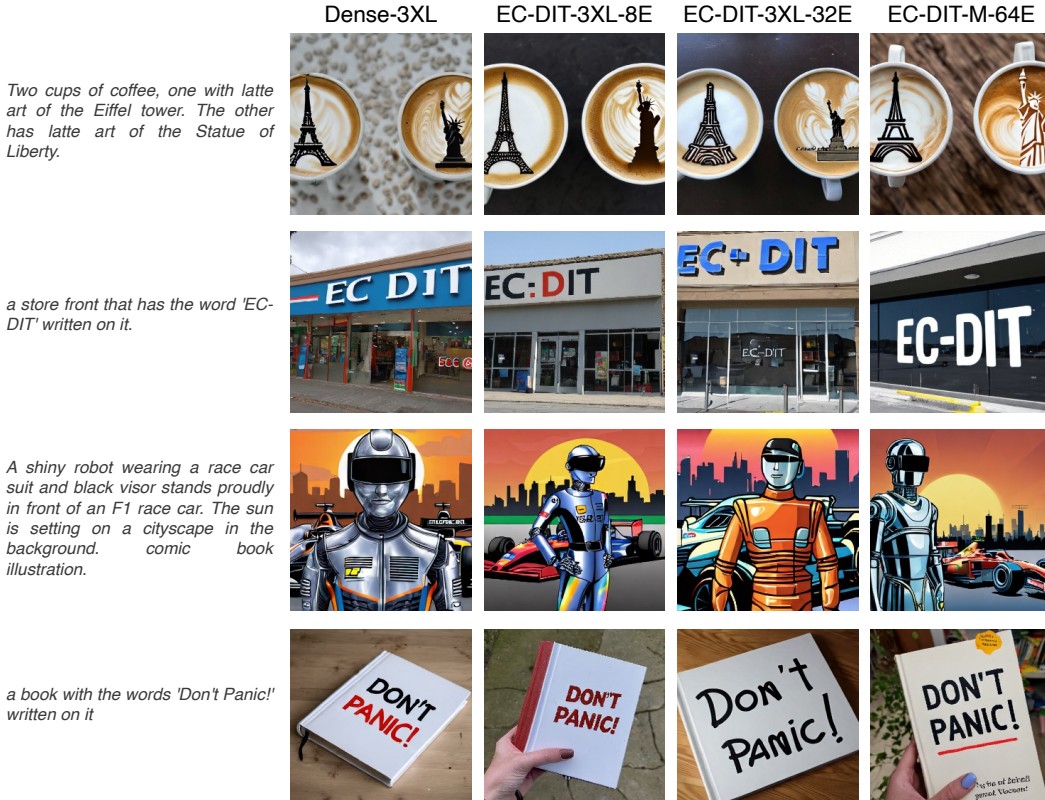

Figure 9: **Visual comparison of three model configurations on the Partiprompts subset**. For each prompt, the model generates four candidates, and the one with the best visual satisfaction is selected.

We provide an additional comparison on the PartiPrompt subset (Yu et al., 2022) across different model configurations and also benchmark against two external text-to-image generative models: SD3-Large (Esser et al., 2024) and FLUX.1[dev] (Black Forest Labs, 2024).

Both SD3-Large and FLUX.1[dev] have comparable or higher activation sizes than EC-DIT-M-64E. Despite this, EC-DIT demonstrates superior text-to-image alignment compared to these models. For example, in Prompt #3, FLUX.1[dev] fails to align with the textual elements `holding a cane` and `holding a garbage bag` simultaneously, nor adhering to the requested style of `abstract cubism`. In contrast, EC-DIT effectively captures both the described actions and the specified artistic style. Similarly, in Prompt #5, SD3-Large and FLUX.1[dev] fail to include the background detail of `Dois Irmãos in Rio de Janeiro`, while EC-DIT correctly incorporates the described view.

Furthermore, EC-DIT generates images with lower hallucination. For instance, in Prompt #2, SD3-Large hallucinates a third claw on the hamster, and FLUX.1[dev] adds an unnecessary extra `"!"` mark. In contrast, EC-DIT 's generation appears visually accurate and reasonable.

EC-DIT-M-64E
(8B)

SD3-Large
(8B)

FLUX.1 [dev]
(12B)

**Prompt #1**

*This dreamlike digital art captures a vibrant, kaleidoscopic bird in a lush rainforest*

**Prompt #2**

*A high contrast portrait photo of a fluffy hamster wearing an orange beanie and sunglasses holding a sign that says 'Ec-dit'*

**Prompt #3**

*A raccoon wearing formal clothes, wearing a top hat and holding a cane. The raccoon is holding a garbage bag. Oil painting in the style of abstract cubism*

**Prompt #4**

*a portrait of a statue of the Egyptian god Anubis wearing aviator goggles, white t-shirt and leather jacket. The city of Los Angeles is in the background.*

**Prompt #5**

*A teddy bear wearing a motorcycle helmet and cape is riding a motorcycle in Rio de Janeiro with Dois Irmãos in the background*

**Prompt #6**

*A cozy living room with a painting of a corgi on the wall above a couch and a round coffee table in front of a couch and a vase of flowers on a coffee table*

**Prompt #7**

*Detailed pen and ink drawing of a happy pig butcher selling meat in its shop*

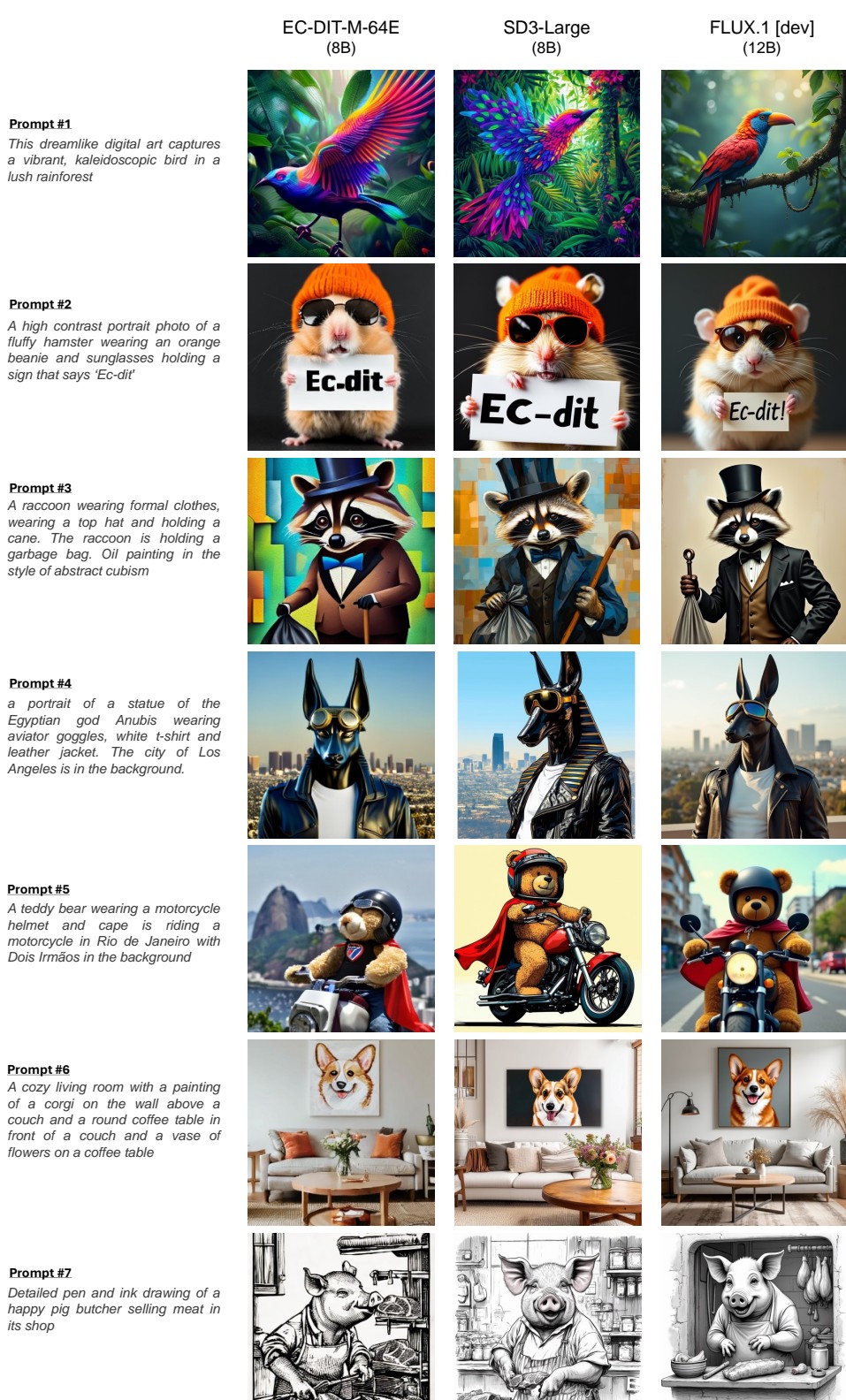

Figure 10: **Visual comparison of EC-DIT-M-64E and two external text-to-image models on the Partiprompts subset**. Numbers in parentheses indicate the activated parameter size for each model. All models leverage rectified flow for generation. SD3-Large inherently generates at $1024 \times 1024$, while FLUX.1[dev] generates at $512 \times 512$.

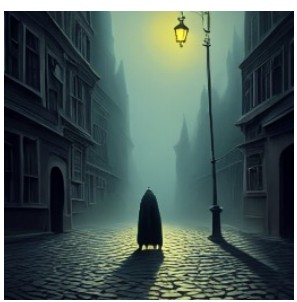

*A solitary figure shrouded in mists peers up from the cobblestone street at the imposing and dark gothic buildings surrounding it. An old-fashioned lamp shines nearby. Oil painting*

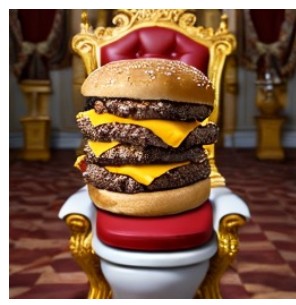

*A cheeseburger with juicy beef patties and melted cheese sits on top of a toilet that looks like a throne and stands in the middle of the royal chamber*

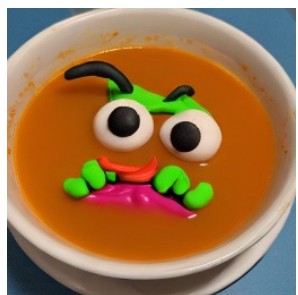

*A bowl of soup that looks like a monster made out of plasticine*

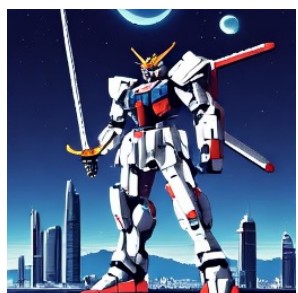

*A gundam stands tall with its sword raised. A city with tall skyscrapers is in the distance, with a mountain and ocean in the background. A dark moon is in the sky. Realistic high-contrast anime illustration*

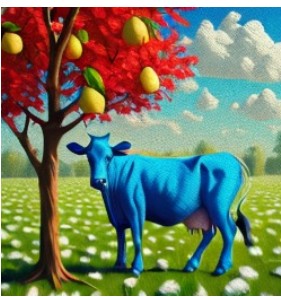

*A blue cow is standing next to a tree with red leaves and yellow fruit. The cow is standing in a field with white flowers. Impressionistic painting*

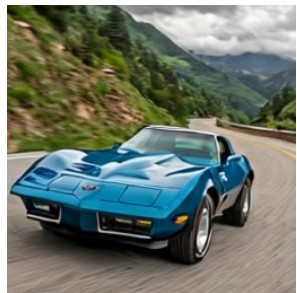

*Three-quarters front view of a blue 1977 Corvette coming around a curve in a mountain road and looking over a green valley on a cloudy day*

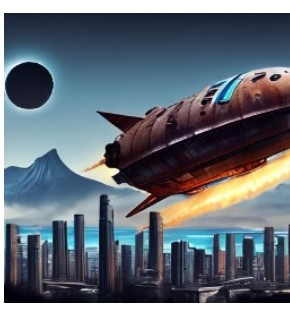

*A rusty spaceship blasts off in the foreground. A city with tall skyscrapers is in the distance, with a mountain and ocean in the background. A dark moon is in the sky. Realistic high-contrast anime illustration*

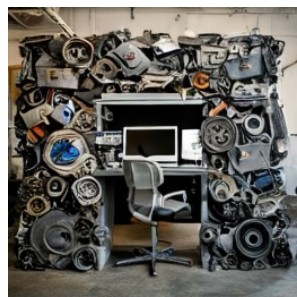

*A small office made out of car parts*

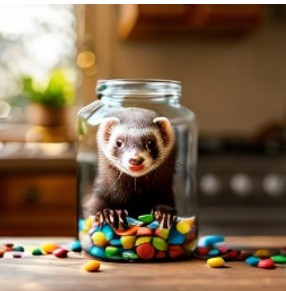

*A mischievous ferret with a playful grin squeezes itself into a large glass jar, surrounded by colorful candy. The jar sits on a wooden table in a cozy kitchen, and warm sunlight filters through a nearby window*

Figure 11: **More generated samples with EC-DɪT-M-64E on the Partiprompts subset**. For each prompt, the model generates four candidates, and the one with the best visual satisfaction is selected.

