# OpenReview forum: "EC-DIT: Scaling Diffusion Transformers with Adaptive Expert-Choice Routing"
_ICLR.cc/2025/Conference — ICLR 2025 Poster_

### Official Review · Reviewer_U4nK · 2024-10-27

**Soundness:** 2
**Presentation:** 3
**Contribution:** 2
**Rating:** 6
**Confidence:** 4

**Summary:**

The paper presents EC-DIT, a new Mixture-of-Experts (MoE) model specifically designed for diffusion transformers in text-to-image synthesis. EC-DIT leverages computational heterogeneity through expert-choice routing, which enables the model to allocate computational resources adaptively based on the complexity of the input text and the corresponding image patches. This approach supports scaling up to 97 billion parameters, leading to improvements in text-to-image alignment, and generation quality.

**Strengths:**

1. Investigating methods to scale up diffusion transformers is a crucial research direction, as it directly impacts the effectiveness and potential of large-scale text-to-image synthesis models.

2. The proposed EC-DIT model's effectiveness is verified by quantitative metrics.

**Weaknesses:**

1. As shown in Figures 1 and 9, the generated images are relatively simple and lack sufficient complexity. For more intricate images, the aesthetic quality appears subpar, and the benefits of scaling up are not convincingly perceptible. Offering 64-128 more generated images with more complex scenes could better demonstrate the value of the proposed method.

2. Models like SD3 and Flux, which have significantly fewer parameters than EC-DIT-64E, manage to produce images with notably superior visual quality. Please clarify why this discrepancy exists and address what specific advantages scaling up provides if the visual quality is still lacking. This raises questions about the true value that scaling up offers in terms of image quality.

**Questions:**

Please refer to the weaknesses.

---

> ### Author Response · Authors · 2024-11-21
>
> We sincerely appreciate the Reviewer for providing valuable feedback. Please find the responses below:
>
> >   As shown in Figures 1 and 9, the generated images are relatively simple and lack sufficient complexity. For more intricate images, the aesthetic quality appears subpar, and the benefits of scaling up are not convincingly perceptible. Offering 64-128 more generated images with more complex scenes could better demonstrate the value of the proposed method.
>
> **Response:**  Following the Reviewer’s suggestion, we have generated an additional set of 16 samples using EC-DiT-M-64E, exploring a variety of topics and art styles, as detailed in the revised Figures 9 and 10 in Appendix E. For a more comprehensive comparison, we included several prompts from the SD3 paper [1] (e.g., prompts from Figure 12 in [1]). These additional samples demonstrate that EC-DiT can produce visually appealing results across diverse domains and styles. In Figures 1 and 9 of the original draft, we aimed to showcase how scaling with the EC-DiT framework enhances the dense model’s generation performance, including improvements in text-to-image alignment and visual detail quality.
>
> Moreover, the primary focus of our work is the context-aware adaptive computation framework designed to efficiently scale text-to-image diffusion models. It is worth noting that direct comparisons with external models regarding scalability and efficiency are challenging, as many existing models are only partially open-sourced or entirely closed-sourced. To address this, we emphasize comparisons between EC-DiT and dense or token-choice baselines, as presented in Figures 3-5 and 7. These models were trained under consistent setups, ensuring a fair and reliable evaluation of our method’s scalability and efficiency.
>
> >  Models like SD3 and Flux, which have significantly fewer parameters than EC-DIT-64E, manage to produce images with notably superior visual quality. Please clarify why this discrepancy exists and address what specific advantages scaling up provides if the visual quality is still lacking. This raises questions about the true value that scaling up offers in terms of image quality.
>
> **Response:**  Thank you for the feedback. Models like SD3 and Flux are trained or fine-tuned at higher resolutions, such as $512^2$ and $1024^2$, which include more intricate visual patterns and positively impact metrics like GenEval scores (as noted in Table 5 of [4]). Since these techniques are orthogonal to our scaling method, we pretrained EC-DiT at a resolution of $256^2$ without employing DPO fine-tuning. These differences in resolution and fine-tuning approaches partly explain the discrepancy in visual quality.
>
> Despite these differences, our results demonstrate that EC-DiT surpasses these baselines in GenEval metrics, even without additional fine-tuning or higher resolutions. For instance, as shown in Table 2, EC-DiT achieves a GenEval score of 71.68%, compared to SD3 (8B)’s score of 68.00%, even though both models have a comparable number of activated parameters. Furthermore, EC-DiT outperforms SD3 with DPO fine-tuning, which achieves a score of 71.00%. These results highlight the effectiveness of our scaling framework, even in the absence of additional resolution or fine-tuning enhancements.
>
> Additionally, EC-DiT achieves comparable performance to state-of-the-art models with significantly fewer activated parameters. For example, EC-DiT-3XL-32E, with 5.18B activated parameters (64% of SD3-Large), achieves a nearly equivalent GenEval score of 70.91% compared to SD3-Large’s 71.00%. Notably, EC-DiT-3XL-32E’s inference time is only 33.8% of the 8B dense variant (Dense-M), further emphasizing the efficiency of our approach.
>
> It is also worth noting that the datasets used for pretraining also differ between EC-DiT and external models like SD3 and Flux. As previously discussed, direct comparisons of visual quality alone may not fully represent the effectiveness of different approaches. Thus, we have conducted extensive comparisons between EC-DiT and dense or token-choice baselines, as shown in Figures 3-5 and 7. These comparisons highlight how our scaling approach enhances both training and inference performance. When applied to the same baseline, EC-DiT consistently improves visual quality and textual alignment.
>
> We appreciate the Reviewer’s feedback and are actively conducting further studies to strengthen our evaluation. These include training on publicly available datasets used by SD3 and performing super-resolution fine-tuning to better showcase the scalability and potential of our method.
>
>
> **Reference:**
>
> [1] Patrick Esser, Sumith Kulal, Andreas Blattmann, Rahim Entezari, Jonas Muller, Harry Saini, Yam Levi, Dominik Lorenz, Axel Sauer, Frederic Boesel, Dustin Podell, Tim Dock horn, Zion English, Kyle Lacey, Alex Goodwin,Yannik Marek, and Robin Rombach. Scaling rectified flow transformers for high-resolution image synthesis, 2024.

---

> > ### Comment · Reviewer_U4nK · 2024-11-23
> >
> > Thank you for the detailed response and the additional experimental results. To provide a more direct comparison of EC-DiT with existing models such as SD3 and Flux in complex scenes and artistic styles, could you consider supplementing the new Figure 9 with direct comparative generation examples? For instance, showcasing results from EC-DiT, SD3, and Flux on the same prompts involving complex scenes or specific artistic styles would help illustrate the visual quality and text alignment advantages of EC-DiT more concretely.

---

> > > ### Author Response · Authors · 2024-11-24
> > >
> > > Thank you for the follow-ups. We have updated **Figure 10, Appendix E** in our paper with a side-by-side comparison of EC-DiT and the external models suggested. For SD3, we selected its largest model, SD3-Large, which has 8B activated parameters and generates images at a resolution of $1024^2$. For FLUX, we chose their best-performing open-weight model, FLUX.1[dev], which features 12B parameters and a resolution of $512^2$. To ensure optimal generation, we used their official APIs/playgrounds [2] and [3], importing all default parameters and inputting only the prompts being tested.
> > >
> > > As illustrated in the new Figure 10 of the revised draft, EC-DiT demonstrates **superior text-to-image alignment** compared to these models. For instance:
> > >
> > > -  In Prompt #3, FLUX.1[dev] fails to align with the textual elements _holding a cane_ and _holding a garbage bag_ simultaneously, nor adhering to the requested style of _abstract cubism_. In contrast, EC-DiT effectively captures both the described actions and the specified artistic style.
> > >
> > > -  In Prompt #5, SD3-Large and FLUX.1[dev] omit the background detail of _Dois Irmãos in Rio de Janeiro_, while EC-DiT successfully incorporates the described view.
> > >
> > > Additionally, EC-DiT produces images with **fewer hallucinations**. For example:
> > >
> > > -  In Prompt #2, SD3-Large hallucinates a third claw on the hamster, and FLUX.1[dev] introduces an unnecessary extra _”!”_ mark. In contrast, EC-DiT’s generation appears visually accurate and reasonable.
> > >
> > > We acknowledge that in certain cases, such as Prompt #2 and Prompt #7, SD3-Large and FLUX.1[dev] generate smoother images compared to EC-DiT. This may be attributed to their training on higher resolutions or larger activated parameters. We are actively conducting further studies, including super-resolution fine-tuning of EC-DiT, to better demonstrate the scalability and potential of our method.
> > >
> > > **Reference:**
> > >
> > > [2] https://platform.stability.ai/docs/api-reference
> > >
> > > [3] https://fal.ai/models/fal-ai/flux/dev/playground

---

> > > > ### Comment · Reviewer_U4nK · 2024-11-26
> > > >
> > > > Thank you for sharing the updated experiments and insights. Overall, EC-DiT demonstrates promising performance, achieving results comparable to SD3-Large and FLUX.1[dev]. However, although it has a total parameter number of 97B, it still faces limitations in detailed generation and overall aesthetic quality. For the details, I suspect the primary constraint is the resolution being set at 256x256. On the aesthetic side, the data used for training may also play a significant role. Considering the extension of the rebuttal discussion period by one week, I encourage you to explore fine-tuning the model on a small subset of high-quality 512x512 resolution data. Validating the performance using the same prompts as in Figure 10 could provide valuable insights. I believe such additional experiments could further demonstrate the scalability and potential of EC-DiT.

---

> ### Author Response · Authors · 2024-11-25
>
> Dear Reviewer U4nK,
>
> Thank you once again for your insightful feedback and suggestions. As the discussion period draws to a close, we kindly invite you to review our newly added results and discussions, both in the responses above and the revised draft.
>
> We greatly appreciate the time and effort you have devoted to reviewing our work. Should there be any remaining points you would like us to clarify or address further, please do not hesitate to let us know.
>
> Best regards,
>
> The Authors

---

> ### Author Response · Authors · 2024-11-29
>
> Thank you for the suggestion. We agree that super-resolution fine-tuning would further demonstrate the scalability and potential of EC-DiT, and we are actively conducting studies in this direction. However, as stated in the adjustments to the rebuttal discussion period, it is not permitted to update the PDF with additional experimental results after the original rebuttal deadline (11/27). To further strengthen the evaluation of our method, we commit to releasing our model weights and including the fine-tuning results upon acceptance.
>
> Additionally, we want to emphasize that our primary contribution is the **context-aware adaptive computation framework** designed to efficiently scale text-to-image (T2I) generative models, which is *orthogonal* to super-resolution fine-tuning. We have provided extensive evaluations and analyses that highlight the advantages of our framework over existing dense and token-choice approaches:
>
> - **Outperforming T2I Alignment**:
> As shown in Table 2, our largest model, even without additional fine-tuning, surpasses the SD3 baseline with DPO fine-tuning and higher resolution. Furthermore, our 3XL model, of which the activated size is 36% smaller than the SD3-Large baseline, achieves comparable performance, demonstrating the efficiency of our scaling method. The superior alignment of EC-DiT is also qualitatively evident in the newly added Figure 10, as discussed in our previous response.
>
> - **Efficient Scaling**:
> Scaling dense models with EC-DiT consistently delivers superior performance over dense and token-choice sparse DiT approaches across multiple dimensions, including text-to-image alignment (Figures 4 and 7), image generation quality (Figure 7), and training convergence (Figure 8).
>
> - **Adaptive Computation Allocation**:
> Visualizations in Figure 6 further illustrate that EC-DiT effectively learns adaptive computation allocation that aligns strongly with the textual significance of tokens.
>
> We would like to note that our experimental setups and evaluations are **self-contained**. As the Reviewer suggested, training data can significantly affect the overall performance. Thus, to ensure a fair comparison, we conducted all training and evaluation of both EC-DiT and the dense/token-choice baselines with the same training data, resolution, backbone architecture. Therefore, we believe these results sufficiently demonstrate EC-DiT’s promising scalability and effectiveness.
>
> We greatly appreciate the Reviewer’s discussion and suggestions. If there are any additional points we can clarify or address further, please feel free to let us know.

---

> > ### Comment · Reviewer_U4nK · 2024-11-29
> >
> > Thank you for the detailed reply. I’m pleased to see the efforts in adding experiments and the commitment to releasing model weights. Based on this, I am happy to increase my score from 5 to 6. Looking forward to seeing more exciting results.

---

> > > ### Author Response · Authors · 2024-11-29
> > >
> > > Thank you for your thoughtful feedback and for increasing the score. We look forward to sharing more detailed results in the future.

---

### Official Review · Reviewer_AMiW · 2024-10-30

**Soundness:** 3
**Presentation:** 3
**Contribution:** 3
**Rating:** 6
**Confidence:** 3

**Summary:**

This paper explores how to scale diffusion transformers (DiT) to larger model capacities. The authors developed EC-DiT, a MoE-based (Mixture of Experts) DiT model leveraging expert-choice routing, expanding the DiT model parameters to 97 billion. EC-DiT enables context-aware routing and dynamically adjusts computation, inherently ensuring load balancing across experts. Extensive experiments were conducted to verify and analyze the effectiveness of EC-DiT.

**Strengths:**

- EC-DiT successfully scales the DiT model to 97 billion parameters, demonstrating exceptional performance.
- Expert-choice routing outperforms traditional token-choice routing strategies while maintaining competitive inference speed.
- The paper presents extensive experiments to showcase EC-DiT’s capabilities.

**Weaknesses:**

- While I recognize the authors’ contribution in scaling up DiT and support the paper’s acceptance, I believe it does not demonstrate sufficient technical innovation beyond prior work, appearing more as a practical integration of existing techniques
- A clearer and more precise definition of ‘Activated Params.’ should be provided. In Algorithm 1, it appears that all parameters are activated. Does this refer to the average number of parameters activated per token?
- I would like the authors to include a comparison of memory usage during training and inference.
- It would be beneficial if the authors could open-source the relevant scripts and model weights upon acceptance.

**Questions:**

Please refer to the weaknesses part.

---

> ### Author Response · Authors · 2024-11-21
>
> We would like to thank the Reviewer for the insightful feedback. Please find our responses below:
>
> >   While I recognize the authors’ contribution in scaling up DiT and support the paper’s acceptance, I believe it does not demonstrate sufficient technical innovation beyond prior work, appearing more as a practical integration of existing techniques.
>
> **Response:**  Thank you for recognizing the contribution of our work in scaling up DiT models. Our primary contribution is the context-aware adaptive computation framework designed to efficiently scale text-to-image (T2I) generative models. As discussed in Section 4, existing approaches [1-3] for scaling T2I models with MoE often face challenges such as ineffective use of global image information, load imbalance issues, or limited scaling studies.
>
> In contrast, our method achieves adaptive compute allocation and enables the scalability of DiT models to a much larger scale. As presented in Section 3, our approach improved both scalability and  generation performance compared to existing methods for scaling image generative models. To the best of our knowledge, this work is the first to employ heterogeneous computation within large-scale T2I generative models scaling up to 97B parameters.
>
> Additionally, we provide extensive evaluations and analyses that underscore the advantages of our framework over existing dense and token-choice approaches:
>
> - Outperforming T2I Alignment: As shown in Table 2, our largest model, even without additional fine-tuning, surpasses the SD3 baseline that benefits from DPO fine-tuning and higher resolution. Furthermore, our 3XL model, which is 36% smaller than the SD3-Large baseline, achieves comparable performance, showcasing the efficiency of our scaling method.
>
> - Efficient Scaling: Scaling dense models with EC-DiT consistently delivers superior performance over dense and token-choice sparse DiT approaches across multiple dimensions, such as text-to-image alignment (Figures 4 and 7), image generation quality (Figure 7) and training convergence (Figure 8).
>
> - Visualizations in Figure 6 further demonstrate that the model effectively learns adaptive computation allocation that aligns strongly with the textual significance of tokens.
>
> These results highlight the novelty and effectiveness of our approach. We humbly believe our method goes beyond a practical integration of existing techniques to introduce an efficient scaling methodology for T2I generative models.
>
>
>
> >  A clearer and more precise definition of ‘Activated Params.’ should be provided. In Algorithm 1, it appears that all parameters are activated. Does this refer to the average number of parameters activated per token?
>
> **Response:**  The Activated Params. represent the average number of parameters activated per token. With the expert-choice routing employed in EC-DiT, the capacity factor $f_c$ determines the overall computation. We use the following pseudocode to calculate the increment in activated parameters introduced in sparse layers and supplement it to the dense total parameters to get the Activated Params. in Table 1.
>
> ```python
> # Dense FFN Size
> dense_ffn_size = 2 * hidden_dim * hidden_dim * ffn_factor
>
> # Router Size
> router_size = num_experts * hidden_dim
>
> # Attention Size
> attn_size = hidden_dim * (num_heads + num_kv_heads * 2 + num_heads) * attn_key_dim
>
> # Activated Dense Total
> activated_dense_total = (attn_size + dense_ffn_size) * num_sparse_layers
>
> # Activated Sparse Total
> activated_sparse_total = (
>     attn_size
>     + 2 * hidden_dim * hidden_dim * ffn_factor * capacity_factor
>     + router_size
> ) * num_sparse_layers
>
> # Activated Total Increment
> activated_increment = activated_sparse_total - activated_dense_total
> ```
>
> The parameters `hidden_dim`, `num_sparse_layers`, `num_heads`, and `num_kv_heads` for each model configuration are specified in Table 1. We set `ffn_factor=4.0` and `capacity_factor=2.0` globally.
>
> We have revised the caption for Table 1 and included the pseudocode in Appendix D accordingly.

---

> ### Author Response · Authors · 2024-11-21
>
> >  I would like the authors to include a comparison of memory usage during training and inference.
>
> **Response:**  We summarized the computation resources used during training and inference for each configuration with 32 experts or above.
>
> | Model | Training Specs (TPU version * topology) | Training Time (hour) | Inference Memory (per-GPU memory * number of GPUs) | Inference Parallelism | Activated Params |
> |------------|--------------------------------------------------|--------------------|-------------------------|-----------------------|-------------------|
> | EC-DiT-XL-32E | v4 * (4,4,8) | 75.6 | 12.8GB * 8 | DDP | 1.62B |
> | EC-DiT-XXL-32E | v5p * (4,8,8) | 82.2 | 20.7GB * 8 | DDP | 2.71B |
> | EC-DiT-3XL-32E | v5p * (4,8,8) | 96.1 | 39.6GB * 8 | FSDP | 5.18B |
> | EC-DiT-M-64E | v5p * (8,8,8) | 100 | 60.7GB * 8 | Model | 8.27B |
>
> Here TPU v4 and v5p have 32GB and 95GB memory capacity per chip, respectively.
>
> >  It would be beneficial if the authors could open-source the relevant scripts and model weights upon acceptance.
>
> **Response:**  Thank you for the suggestion. We will open-source our model weights and relevant scripts upon acceptance to support reproducibility.
>
> **Reference:**
>
> [1] Gupta, Y., Jaddipal, V.V., and Prabhala, H. SegMoE: Segmind Mixture of Diffusion Experts.
>
> [2] Park, B., Go, H., Kim, J.-Y., Woo, S., Ham, S., and Kim, C. 2024. Switch Diffusion Transformer: Synergizing Denoising Tasks with Sparse Mixture-of-Experts. arXiv.
>
> [3] Lin, X.V., Shrivastava, A., Luo, L., et al. 2024. MoMa: Efficient Early-Fusion Pre-training with Mixture of Modality-Aware Experts. arXiv.

---

### Official Review · Reviewer_J1KC · 2024-11-04

**Soundness:** 2
**Presentation:** 4
**Contribution:** 2
**Rating:** 6
**Confidence:** 4

**Summary:**

This paper proposes a MoE based diffusion transformer framework to scale diffusion models. The basic idea is to combine DIT and MoE. The experimental results demontrate the effectiveness.

**Strengths:**

i) The technique line is reasonable.
ii) The writing is easy to follow.

**Weaknesses:**

i) This work should be incremental in essence, as the basic idea is to combine DIT and MoE.
ii) Certain statements exhibit overclaims that need refinement. (see Questions in detail)
iii) Crucial experiment comparisons with diffusion methods utilizing MoE are notably absent. (see Questions in detail)

The rebuttal addresses my concerns, so I raise the score.

**Questions:**

While efforts have been made to enhance the scalability of diffusion models for improved performance and efficiency, several issues remain:
i) The novelty of this approach requires clarification, especially in comparison to existing diffusion models incorporating MoE, as detailed in the related work section (lines 057-063).
ii) Certain claims should be refined or revised. For instance, the statement: “our approach significantly improves training convergence, text-to-image alignment, and overall generation quality compared to dense and sparse variants with conventional token-choice MoEs (Lepikhin et al., 2020b; Du et al.,2022; Zoph et al., 2022).” First, I do not see the comparison with these methods in experiments; Second, the performance is only comparable with SOAT, thereby questioning the usage of 'significantly.'; Third, the reason of improvement might be mainly attributed to increased model parameters, e.g., in experience, more parameters make the training converge faster. Thus, deeper analysis could be given.
“EC-DIT leverages this global context, combining multi-modal information to optimize token-to-expert allocation.” multi-modal?
“The model resolution is set to 256 × 256, with a patch size of 2.” “This results in the input sequence length of 128 per image.” Patch size is 2*2?
iii) Those existing diffusion models with MoE should be compared in experiments, if they are available.
iv) Fig. 3 only shows the performance on DSG score. Additional evaluation on other metrics should be showed, as the performance across various metrics might exhibit different trends.
v) Fig. 4 should show more comparisons with other methods (e.g., in Table 2), rather than solely focusing on the proposed methods themselves.
vi) How to set C in Top-k? There lacks the comparison with the different C values.
vii) The performance is only comparable with SOAT methods, such as SD3. In fact, the parameter quantities are in the same magnitude, SD3 (8B) vs EC-DIT-M-64E(>8B, observed in Fig. 4). Thus, this raises questions about the core contributions of this work. Is the primary focus acceleration? If so, there lacks many comparisons to current methods.

---

> ### Author Response · Authors · 2024-11-21
>
> Thank you for the insightful comments. Please find our responses below:
>
> >  This work should be incremental in essence, as the basic idea is to combine DIT and MoE.
>
> **Response:** We would like to highlight our primary contribution as the context-aware adaptive computation framework designed to efficiently scale text-to-image (T2I) generative models. As discussed in Section 4, existing approaches [1-3] to scaling T2I models with MoE often fail to leverage global image information, suffer from load imbalance issues, or conduct limited scaling studies. In contrast, our method achieves adaptive compute allocation with perfect load balance and demonstrates the scalability of DiT models on a much larger scale. As presented in Section 3, our proposed scaling approach enables both promising scalability and improved generative performance compared to existing scaling methods for image generative models. To the best of our knowledge, this work is the first to employ heterogeneous computation within large-scale T2I generative models, scaling up to 97B parameters. Additionally, we provide extensive evaluations and analyses demonstrating the advantages of our proposed scaling framework over existing dense or token-choice approaches.
>
> >  The novelty of this approach requires clarification, especially in comparison to existing diffusion models incorporating MoE, as detailed in the related work section (lines 057-063).
>
> **Response:** For text-to-image generation tasks, existing approaches such as SegMoE [1] and DiT-MoE [2] often employ sparse MoEs with token-choice routing to produce sparse DiTs of up to 16B parameters. Despite technical differences, these methods, in principle, rely on routing text tokens to top-ranked experts, a strategy that is suboptimal for diffusion models. Additionally, token-choice routing frequently suffers from load imbalance, necessitating an auxiliary balancing loss to address this issue.
>
> To showcase the advantages of the proposed EC-DiT, we compared FID and CLIP Score trends with the token-choice approach, as detailed in Figure 5 and Section 3.5. For a fair comparison, we maintained identical experimental settings, except for replacing the routing strategy with token-choice and incorporating an auxiliary load-balancing loss in the training objective. As demonstrated, EC-DiT consistently achieves faster training convergence and better performance throughout training. Notably, EC-DiT with 8 experts rivals the token-choice baseline with 16 experts in both generation quality and text-image alignment. Furthermore, EC-DiT with more experts significantly outperforms the token-choice baseline.
>
> To further evaluate text-to-image alignment, we compared EC-DiT with the token-choice method (denoted as GS) on both GenEval and DSG scores.
>
> > GenEval comparison with token-choice baselines (GS)
>
> | Model (↓) / num_experts (→) | 8E    | 16E    | 32E    |
> |--------------------|-------|--------|--------|
> | **GS-XL (Token-choice)**          | 67.82 | 68.00  | 68.52  |
> | **EC-DiT-XL**          | **68.62** | **68.76**  | **69.38**  |
> | **GS-XXL (Token-choice)**         | 68.30 | 68.49  | 69.96  |
> | **EC-DiT-XXL**         | **69.11** | **69.43**  | **70.86**  |
>
> >DSG comparison with token-choice baselines (GS)
>
> | Model (↓) / num_experts (→) | 8E    | 16E    | 32E    |
> |-----------------------------|-------|--------|--------|
> | **GS-XL (Token-choice)**    | 73.47 | 73.90  | 74.07  |
> | **EC-DiT-XL**               | **74.03** | **74.03**  | **74.58**  |
> | **GS-XXL (Token-choice)**   | **75.10** | 75.30  | 75.40  |
> | **EC-DiT-XXL**              | 75.03 | **75.63**  | **75.93**  |
>
> By leveraging global image information and adaptive computation, EC-DiT demonstrates superior performance over the token-choice baseline across most evaluation categories.
>
> Following the Reviewer’s comment, we have revised our paper to include these detailed results in Tables 3 and 4 of Appendix B for a more comprehensive comparison.

---

> ### Author Response · Authors · 2024-11-21
>
> >  Certain claims should be refined or revised. For instance, the statement: “our approach significantly improves training convergence, text-to-image alignment, and overall generation quality compared to dense and sparse variants with conventional token-choice MoEs (Lepikhin et al., 2020b; Du et al.,2022; Zoph et al., 2022).” First, I do not see the comparison with these methods in experiments;
>
> **Response:**  We apologize for the confusion. The methods mentioned here refer to conventional token-choice MoEs primarily used in language models. For text-to-image generation tasks, existing approaches employing token-choice routing include SegMoE [1] and DiT-MoE [2]. As discussed in our previous response, despite several technical differences, these methods fundamentally rely on routing text tokens to top-ranked experts, which is suboptimal for diffusion models. Furthermore, token-choice routing often encounters load imbalance issues, requiring an auxiliary balancing loss to mitigate this problem.
>
> To highlight the advantages of the proposed EC-DiT, we conducted comparisons of FID and CLIP Score trends against the token-choice approach, as presented in Figure 5 and Section 3.5. Additionally, we provided comparisons of text-to-image alignment performance, as detailed in the last response.
>
> We have revised our paper to include these results for improved clarity and to better support our claims.
>
> >   Second, the performance is only comparable with SOAT, thereby questioning the usage of 'significantly.';
>
> >   The performance is only comparable with SOAT methods, such as SD3. In fact, the parameter quantities are in the same magnitude, SD3 (8B) vs EC-DIT-M-64E(>8B, observed in Fig. 4). Thus, this raises questions about the core contributions of this work. Is the primary focus acceleration? If so, there lacks many comparisons to current methods.
>
>
> **Response:**  The primary focus of our work is on the context-aware adaptive computation framework, designed for efficiently scaling text-to-image diffusion models. In Table 2, we demonstrate that, with our proposed scaling technique, EC-DiT achieves superior performance of 71.68% compared to the state-of-the-art SD3 (8B) of 68.00%, which has a comparable number of activated parameters. To further emphasize the efficiency of our approach, we include SD3 w/ DPO of 71.00% in Table 2 and show that our method can outperform even the fine-tuned variant.
>
> Specifically, EC-DiT-M-64E achieves a GenEval score of 71.68%, pretrained at a resolution of $256\times256$ without DPO fine-tuning. In contrast, the SOTA score for solely pretrained SD3 at a resolution of $512\times512$ is 68.00%, which is notably lower than our result. As outlined in Table 5 of [4], both DPO fine-tuning stage and higher resolution positively impact GenEval scores. Importantly, these techniques are orthogonal to our scaling method, and our results reflect EC-DiT’s performance without fine-tuning at a higher resolution. This demonstrates that even without this additional stage, EC-DiT outperforms the SOTA baselines.
>
> Additionally, SD3-Large, with a GenEval score of 71.00%, is an 8B dense model. In comparison, EC-DiT-3XL-32E, with 5.18B activated parameters (64% of SD3-Large), achieves a nearly equivalent GenEval score of 70.91%. Notably, EC-DiT-3XL-32E’s inference time is only 33.8% of the 8B dense variant (Dense-M). These results highlight that our scaling approach can deliver performance comparable to significantly larger dense models.
>
> Overall, our method, even without an additional fine-tuning stage, surpasses the SOTA baseline that benefits from fine-tuning and higher resolution. Additionally, our method applied to a model that is 36% smaller than the SOTA achieves a similar performance to the SOTA.
>
> We thank the Reviewer for raising this point and have updated our paper to include this discussion. We are also actively working on fine-tuning EC-DiT at higher resolutions to explore further improvements.

---

> ### Author Response · Authors · 2024-11-21
>
> >   Third, the reason of improvement might be mainly attributed to increased model parameters, e.g., in experience, more parameters make the training converge faster. Thus, deeper analysis could be given.
>
> **Response:**  Our evaluation results demonstrate that the proposed EC-DiT leverages global image information to efficiently scale the model’s capacity. To further illustrate this, the following table summarizes four metrics for three models. Here, activated parameters refer to the number of parameters activated during the forward pass, which directly correlates with inference speed.
>
> | Model | GenEval (%) ↑ | DSG (%) ↑ | FID ↓ | CLIP ↑ | Activated paramters |
> |-------------------------------|---------------|------------|------------|-----------|---------------------|
> | EC-DiT-XXL-32E | **70.86** | **75.90** | **15.78** | **0.2855** | **2.7B** |
> | Dense-3XL | 69.92 | 75.00 | 16.82| 0.2842 | 4.5B |
>
> We first compare EC-DiT-XXL-32E with the larger dense model (Dense-3XL). EC-DiT-XXL activates 1.8B fewer parameters than the dense 3XL model while surpassing it in all four metrics. This demonstrates that the proposed context-aware scaling approach enables EC-DiT to achieve better performance with a smaller activated model size.
>
>   | Model | GenEval (%) ↑ | DSG (%) ↑ | FID ↓ | CLIP ↑ | Activated paramters |
> |-------------------------------|---------------|------------|------------|-----------|---------------------|
> | EC-DiT-XXL-32E | **70.86** | **75.90** | **15.78** | **0.2855** | 2.7B |
> | GS-XXL-32E (Token-choice) | 69.96 | 74.83 | 16.43 | 0.2850 | 2.7B |
>
> Next, we compare EC-DiT-XXL-32E with the token-choice baseline (GS-XXL-32E), EC-DiT consistently outperforms the token-choice approach across all metrics, despite having the same activated model size. This highlights the effectiveness of EC routing over token-choice routing.
>
> In general, the performance improvements are not merely a result of increased model parameters. The proposed EC-DiT effectively scales a smaller model to outperform a larger dense model, showcasing its ability to achieve superior results. Additionally, this scaling approach proves to be more effective than conventional MoE techniques in text-to-image synthesis tasks.
>
> >  “EC-DIT leverages this global context, combining multi-modal information to optimize token-to-expert allocation.” multi-modal?
>
> **Response:** We are sorry for the ambiguity. By multi-modal, we meant that EC-DiT takes both textual and visual information from the sequence to efficiently route tokens to each expert. The textual information from the prompt is incorporated into the input to the router via the cross-attention module, as presented in Equation 4. The sequence, which contains both textual and image information, is further processed within the EC router as described in Equations 5 and 6.
>
> We have revised our paper to include a more precise and clear description of this process.
>
> >   “The model resolution is set to 256 × 256, with a patch size of 2.” “This results in the input sequence length of 128 per image.” Patch size is 2*2?
>
> **Response:**  Thank you for pointing this out. The patch size is $2 \times 2$. Given the visual encoder’s downsampling factor of 8 and a loss masking ratio of 0.5, the calculation is $(\frac{256}{8\times2})^2*0.5=128$.
>
> We have revised the corresponding section in the paper to include this clarification.
>
> >   Fig. 3 only shows the performance on DSG score. Additional evaluation on other metrics should be showed, as the performance across various metrics might exhibit different trends.
>
> **Response:**  In the experiments, we evaluated EC-DiT across four base model configurations with varying numbers of experts (ranging from 8 to 64) on several key metrics: GenEval (Table 2 and Figure 4), DSG (Figure 3), FID and CLIP Score (Figure 7), inference time elapsed (Figure 4), and training loss (Figure 8). These metrics collectively validate the effectiveness and efficiency of the proposed EC-DiT in terms of text-to-image alignment, image generation quality, inference speed, and training convergence.
>
> As shown in the results, scaling dense models with EC-DiT consistently demonstrates superior performance across various dimensions, such as text-to-image alignment (Figure 4 and Figure 7), image generation quality (Figure 7), and training dynamics (Figure 8). Furthermore, increasing the number of experts enhances performance while introducing only small inference overhead, which also highlights the scalability and efficiency of the proposed approach.

---

> ### Author Response · Authors · 2024-11-21
>
> >  Fig. 4 should show more comparisons with other methods (e.g., in Table 2), rather than solely focusing on the proposed methods themselves.
>
> **Response:**  Thank you for the suggestion. In Figure 4, we reported the GenEval score and inference time for EC-DiT under various settings. We aimed to incorporate comparisons with other baseline methods presented in Table 2 by locally evaluating these methods to ensure accurate inference time measurements. However, certain models, such as our strongest baseline SD3, are only partially open-sourced. Specifically, SD3 models with sizes over 2B parameters are accessible solely via API, which poses challenges in accurately assessing inference efficiency for all models. We will continue evaluating other open-sourced large-scale models and update our paper with additional results to enhance our evaluation as soon as possible.
>
> Moreover, the primary focus of our work is on the context-aware adaptive computation framework designed for efficient scaling. External models often vary significantly in training data and design, which can make direct comparisons less representative of our method’s effectiveness. For this reason, we would like to emphasize comparisons between EC-DiT and the dense and token-choice baselines, as shown in Figures 3-5 and 7. These models share the same training setup, offering a more reliable and consistent assessment of EC-DiT’s scalability and efficiency.
>
> >  How to set C in Top-k? There lacks the comparison with the different C values.
>
> **Response:**  Here $C$ represents the capacity for each expert and is calculated via $C=S\times f_c/E$. A larger $C$ means each expert processes more tokens, thereby allocating more compute to each token. The capacity factor $f_c$ is equivalent to the average number of experts processing each token. As $C$ is determined by the sequence length $S$ and capacity factor $f_c$ jointly, we conducted comparisons on EC-DiT-XXL-16E using a set of values for $S\in\\{256,512,1024\\}$ and $f_c\in\\{1,2\\}$, resulting in $C$ ranging from 16 to 128. The comparative results, conducted on 8$\times$ H100 (80G), are summarized below.
>
> | $S$ | $f_c$ | $C$ | FID | Inference Time (sec/500 items) |
> |---------|---|----|--------------|-----------|
> | 256 | 1 | 16 | 16.45 | 77.94|
> | 256 | 2 | 32 | 16.39 | 83.18 |
> | 512 | 1 | 32 | 16.35 | 78.54 |
> | 512 | 2 | 64 | 16.31 | 83.30 |
> | 1024 | 1 | 64 | 16.31 | 81.03 |
> | 1024 | 2 | 128| 16.17 | 86.30 |
>
> Overall, better FID can be achieved with a larger capacity as more compute is allocated to generate each sample. Higher capacity factors yield better generation quality at the cost of increased inference time. Theoretically, models with the same $f_c$ should exhibit similar inference speeds regardless of sequence length. However, in practice, larger sequence lengths require more VRAM to fit in. When scaling the sequence length from 256/512 to 1024, we needed to switch from DDP to FSDP to manage memory constraints. This explains the increase in inference time for $S=1024$ compared with $S=256/512$ under the same $f_c$.
>
> Considering these factors, we set $S=512$ with $f_c=2$ to achieve a good balance between generation quality and inference time. For practical cases with abundant computational resources, a larger $C$ can be used during inference to achieve higher image quality.
>
> **Reference:**
>
> [1] Gupta, Y., Jaddipal, V.V., and Prabhala, H. SegMoE: Segmind Mixture of Diffusion Experts.
>
> [2] Park, B., Go, H., Kim, J.-Y., Woo, S., Ham, S., and Kim, C. 2024. Switch Diffusion Transformer: Synergizing Denoising Tasks with Sparse Mixture-of-Experts. arXiv.
>
> [3] Lin, X.V., Shrivastava, A., Luo, L., et al. 2024. MoMa: Efficient Early-Fusion Pre-training with Mixture of Modality-Aware Experts. arXiv.
>
> [4] Patrick Esser, Sumith Kulal, Andreas Blattmann, Rahim Entezari, Jonas Muller, Harry Saini, Yam Levi, Dominik Lorenz, Axel Sauer, Frederic Boesel, Dustin Podell, Tim Dock horn, Zion English, Kyle Lacey, Alex Goodwin,Yannik Marek, and Robin Rombach. Scaling rectified flow transformers for high-resolution image synthesis, 2024.

---

> ### Author Response · Authors · 2024-11-25
>
> Dear Reviewer J1KC,
>
> Thank you once again for your valuable feedback. As the discussion period approaches its conclusion, we kindly invite you to review our responses and the newly added results. We have made every effort to thoroughly address your concerns and incorporate detailed insights based on your suggestions, both in our responses (above) and the revised draft.
>
> We deeply appreciate the time and effort you have dedicated to reviewing our work. If there are any remaining points you would like us to clarify or address further, please do not hesitate to let us know.
>
> Best regards,
>
> The Authors

---

### Official Review · Reviewer_MJns · 2024-11-04

**Soundness:** 3
**Presentation:** 3
**Contribution:** 3
**Rating:** 6
**Confidence:** 4

**Summary:**

This paper focuses on scaling diffusion transformers for text-to-image synthesis. It proposes EC-DIT, a new family of Mixture-of-Experts (MoE) models with adaptive expert-choice routing. By leveraging the computational heterogeneity of image generation, EC-DIT adaptively allocates compute to understand input texts and generate image patches, enabling heterogeneous computation aligned with text-image complexity. It scales up to 97 billion parameters and shows significant improvements in training convergence, text-to-image alignment, and overall generation quality compared to dense and conventional MoE models. Its contributions include introducing EC-DIT for adaptive computation in text-to-image synthesis, achieving promising performance at scale with faster loss convergence and better quality, and conducting comprehensive experiments to validate its superiority and demonstrate effective adaptive compute allocation.

**Strengths:**

1. EC-DIT can scale up to 97 billion parameters, achieving a state-of-the-art GenEval score of 71.68% in text-to-image synthesis.
2. The adaptive expert-choice routing is a novel design. It enables efficient compute allocation by having experts select relevant tokens, ensuring load balance without an extra load-balancing loss. This approach better utilizes computational resources, adapting to the complexity of different image regions and text, and improving overall performance and efficiency in handling inputs.
3. By embedding timestep and using cross-attention modules, EC-DIT effectively integrates global information, which allows the model to understand the context of the image and its relationship with the text, guiding compute allocation at different stages. It results in hierarchical processing, enhancing image quality and text-to-image alignment, and making the generation process more context-aware.

**Weaknesses:**

1. The best score for SD3 (Esser et al., 2024) w/ DPO is 74% in the original paper, which is actually higher than that of EC-DIT (71.68%). Even with the value 71% listed in Table 2,  EC-DIT is only gaining marginally. Given its additional overhead, it is hard to see the value of the gain.
2. The difference between the theoretical and actual inference overhead (e.g., for EC-DIT-M, the theoretical is around 3% but the actual is 23%) indicates that there may be room for optimization in the inference process. The authors may want to explore the factors contributing to this overhead or propose specific strategies for mitigating it, which could further enhance the practical applicability of the model, especially for applications where real-time or fast inference is crucial.
3. The generalizability of the adaptive expert-choice routing is unclear. The performance of EC-DIT is mainly demonstrated in the context of text-to-image synthesis. How would it fit into the task of pre-training mixed-modal, early-fusion language models? It would be interesting to explore its effectiveness and efficiency in other related tasks.
4. This paper could benefit from more detailed ablation studies on specific components of the proposed model. For example, understanding the individual contributions of the timestep embedding, cross-attention modules, and the expert-choice routing mechanism in more detail. This lack of in-depth analysis makes it harder to understand which parts are truly driving its performance.

**Questions:**

see weakness

---

> ### Author Response · Authors · 2024-11-21
>
> We appreciate the Reviewer for providing insightful comments. Please find the responses below:
>
> >  The best score for SD3 (Esser et al., 2024) w/ DPO is 74% in the original paper, which is actually higher than that of EC-DIT (71.68%). Even with the value 71% listed in Table 2, EC-DIT is only gaining marginally. Given its additional overhead, it is hard to see the value of the gain.
>
> **Response:** Thank you for your detailed feedback. The primary focus of our work is on the context-aware adaptive computation framework, which is designed for efficient scaling. In Table 2, we demonstrate that with our proposed scaling technique, EC-DiT achieves superior performance compared to the state-of-the-art SD3 (8B), which has a comparable number of activated parameters. To further highlight the efficiency of our approach, we include SD3 w/ DPO in Table 2 and show that our method can outperform even the fine-tuned variant.
>
> It is worth noting that the 74% GenEval score for SD3 is from a model operating at $1024^2$ resolution and fine-tuned with DPO. As outlined in Table 5 of [1], both DPO fine-tuning and higher resolution positively impact the GenEval score. However, these techniques are orthogonal to our scaling method, and we reported the EC-DiT results based on a resolution of $256^2$ without fine-tuning.
>
> Additionally, SD3-Large, with a GenEval of 71%, is an 8B dense model. In comparison, EC-DiT-M-64E has 8.27B activated parameters, a comparable size with only a marginal theoretical computational overhead (~3%). Furthermore, EC-DiT-3XL-32E, with 5.18B activated parameters, achieves a nearly equivalent GenEval score of 70.91% compared to the 71% score from SD3-Large, with only 64% activation size of SD3-Large. The EC-DiT-3XL-32E’s inference time is only 33.8% of the 8B dense variant (Dense-M). These results demonstrate that our scaling approach can achieve performance comparable to significantly larger dense models, even without additional fine-tuning.
>
> Finally, we emphasize that external models vary significantly in training data and design, making direct comparisons challenging. Thus, we also highlight comparisons between EC-DiT and dense and token-choice baselines in Figures 3-5 and 7. These models share the same training setup, providing a more consistent and reliable evaluation of our approach’s scalability and efficiency.
>
> Following the Reviewer’s suggestion, we have revised the paper with additional discussion. We are also actively working on fine-tuning EC-DiT at higher resolutions to explore the possibility of surpassing the 74% GenEval score.
>
>
>
> >   The difference between the theoretical and actual inference overhead (e.g., for EC-DIT-M, the theoretical is around 3% but the actual is 23%) indicates that there may be room for optimization in the inference process. The authors may want to explore the factors contributing to this overhead or propose specific strategies for mitigating it, which could further enhance the practical applicability of the model, especially for applications where real-time or fast inference is crucial.
>
> **Response:** Thank you for the suggestion. To clarify, we define theoretical overhead as the additional activation parameters EC-DiT requires compared to its dense counterpart. As shown in Table 1, the largest theoretical overhead is approximately 15%. Conversely, actual overhead refers to the increase in inference time for EC-DiT compared to the dense model. The largest discrepancy between theoretical and actual overhead occurs for EC-DiT-M, where the theoretical overhead is around 3%, but the actual overhead is 23%. This discrepancy likely arises because all models are served on 8$\times$H100 GPUs with 80GB VRAM each. Due to resource limitations, we employ model parallelism for EC-DiT-M-64E, which incurs significant cross-GPU communication during each model forward pass. In contrast, the dense model is served using Fully Sharded Data Parallel (FSDP), which reduces communication costs.
>
> We acknowledge that there is substantial room for improving our model’s inference speed. We aim to provide a tradeoff between inference speed and available compute resources. For instance, allocating more GPUs during inference allows larger scaled models to be served with FSDP or DDP, which can significantly reduce inference time. Following the Reviewer’s suggestion, we will explore more efficient configurations under various computational constraints and investigate techniques to further reduce model size while scaling up.

---

> ### Author Response · Authors · 2024-11-21
>
> >   The generalizability of the adaptive expert-choice routing is unclear. The performance of EC-DIT is mainly demonstrated in the context of text-to-image synthesis. How would it fit into the task of pre-training mixed-modal, early-fusion language models? It would be interesting to explore its effectiveness and efficiency in other related tasks.
>
> **Response:** We appreciate this constructive suggestion. The scaling principle of EC-DiT is to increase model capacity via MoE in the FFN and expert-choice routing, which leverages global information across the entire sequence. This adaptive routing mechanism can seamlessly integrate into any sequence- or chunk-wise encoding/generation process. For instance, the adaptive EC routing can be directly applied to the tokenization [2] commonly used in mixed-modal, early-fusion language models like Chameleon [3]. Similarly, in multi-modal models unifying autoregressive and diffusion paradigms (e.g., TransFusion [4]), EC routing can be naturally integrated into the diffusion-based image generation component. Furthermore, EC-DiT could potentially enhance long-video generation methods [5-6], where the global context within each video chunk could be effectively utilized. We are grateful to the Reviewer for highlighting the potential of incorporating EC-DiT into these methods and tasks. We will explore these exciting directions in our future work.
>
> **Reference:**
>
> [1] Patrick Esser, Sumith Kulal, Andreas Blattmann, Rahim Entezari, Jonas Muller, Harry Saini, Yam Levi, Dominik Lorenz, Axel Sauer, Frederic Boesel, Dustin Podell, Tim Dock horn, Zion English, Kyle Lacey, Alex Goodwin,Yannik Marek, and Robin Rombach. Scaling rectified flow transformers for high-resolution image synthesis, 2024.
>
> [2] Oran Gafni, Adam Polyak, Oron Ashual, Shelly Sheynin, Devi Parikh, and Yaniv Taigman. Make-a-scene: Scene-based text-to-image generation with human priors. arXiv preprint arXiv:2203.13131, 2022.
>
> [3] Chameleon Team. (2024). Chameleon: Mixed-Modal Early-Fusion Foundation Models. ArXiv, abs/2405.09818.
>
> [4] Zhou, C., Yu, L., Babu, A., Tirumala, K., Yasunaga, M., Shamis, L., Kahn, J., Ma, X., Zettlemoyer, L. and Levy, O., 2024. Transfusion: Predict the next token and diffuse images with one multi-modal model. arXiv preprint arXiv:2408.11039.
>
> [5] W. Chen, Y. Ji, J. Wu, H. Wu, P. Xie, J. Li, X. Xia, X. Xiao, and L. Lin, “Control-a-video: Controllable text-to-video generation with diffusion models,” arXiv preprint arXiv:2305.13840, 2023.
>
> [6] Y. Guo, C. Yang, A. Rao, M. Agrawala, D. Lin, and B. Dai, “Sparsectrl: Adding sparse controls to text-to-video diffusion models,” arXiv preprint arXiv:2311.16933, 2023.

---

> ### Comment · Reviewer_MJns · 2024-11-24
>
> Thanks the authors for the detailed rebuttal. I am in general satisfied with the responses.

---

> > ### Author Response · Authors · 2024-11-24
> >
> > Thank you very much for taking the time to review our rebuttal and offering insightful feedback.

---

### Official Review · Reviewer_gySw · 2024-11-06

**Soundness:** 3
**Presentation:** 3
**Contribution:** 3
**Rating:** 6
**Confidence:** 4

**Summary:**

This paper proposes a Mixture-of-Experts (MoE) model, called EC-DIT, for diffusion transformers. By applying the prior work of expert-choice routing to Diffusion Transformer (DiT), EC-DIT allocates a fixed amount of tokens to each expert. As the authors suggest, the method has the advantage that tokens on the more complex parts of images can be assigned to more experts, resulting in higher computation allocated to more important tokens and thus higher efficiency.

**Strengths:**

- The proposed method is well-motivated. An image typically contains regions of different complexity, which require different computation costs during generation. It would be beneficial to use the MoE method to allocate computation to different image regions adaptively. In Figure 6, the visualization of the number of experts for each token shows that more experts are used for the image tokens corresponding to the objects (compared to the background) on the images. The figure provides good support for the motivation of this work.
- The quantitative scores of EC-DIT show its effectiveness (Table 2). The evaluation includes multiple metrics, including GenEval, Davidsonian Scene Graph (DSG) scores, FID, and the CLIP Score.
- The paper is well-written and the method is clearly explained with a pseudocode provided.

**Weaknesses:**

- For Table 2, can the authors include the FID metric to compare the generated images' visual quality across the models?
- Can the authors add a column for inference time or throughput (e.g. images/second) in Table 2? It would help the readers better understand the efficiency of different models.
- This paper has a good contribution to the community by showing the benefits of applying expert-choice routing in DiT, although the main approaches are mainly based on existing works, which makes the proposed method slightly less novel.
- Can the authors make the notations in Section 2.1 regarding the rectified flow clearer? Please explain $p_0$, $x_0$, $x_t$, and $v_\theta$ in more details.

**Questions:**

- We know that DiT performs the diffusion in the latent space instead of the actual image pixel space. Does it undermine the motivation of this paper to allocate more computations to image regions of higher complexity? Please explain how working in latent space affects their approach to computational allocation.
- From Section 2.1, it seems that the authors use the Rectified Flow method on the DiT architecture with MoE. Can the authors explain why they chose to use Rectified Flow and how it integrates with their overall approach? This would help clarify the role of Rectified Flow in their method.

---

> ### Author Response · Authors · 2024-11-21
>
> We would like to thank the Reviewer for the constructive feedback. Please find our responses below:
>
> >   For Table 2, can the authors include the FID metric to compare the generated images' visual quality across the models? Can the authors add a column for inference time or throughput (e.g. images/second) in Table 2? It would help the readers better understand the efficiency of different models.
>
> **Response:** Thank you for the suggestion. The baseline performance data in Table 2 is derived from the SD3 paper [1], which only includes GenEval scores. We are currently working on locally evaluating these baselines to provide additional metrics, including FID and inference throughput. However, some models, such as SD3, are only partially open-sourced—the models of size over 2B are accessible solely via API. This presents challenges in accurately assessing inference efficiency across all models. We will continue to work on cross-model comparisons for open-sourced models and will update our paper with additional results to enhance our evaluation once finished.
>
> Moreover, the primary focus of our work is the context-aware adaptive computation framework designed for efficient scaling up. Our method is theoretically applicable to most DiT-based generative models, including our strongest baseline, SD3 with the MM-DiT architecture. Since external models vary significantly in training data and design, direct comparisons may not accurately represent the effectiveness of our proposed scaling approach. Therefore, we also emphasize comparisons between EC-DiT and the dense and token-choice baselines, as presented in Figures 3-5 and 7. These models share the same training setup and thus offer a more reliable assessment of our approach’s scalability and efficiency.
>
> >   This paper has a good contribution to the community by showing the benefits of applying expert-choice routing in DiT, although the main approaches are mainly based on existing works, which makes the proposed method slightly less novel.
>
> **Response:**  Thank you for the feedback. Our primary contribution lies in introducing adaptive computation to more effectively harness global context information tailored specifically for text-to-image (t2i) generation. This leads to significant scalability and improved generative performance over existing scaling methods in t2i models. To our knowledge, this work is the first to apply heterogeneous computation within large-scale t2i generative models, achieving scalability up to 80B parameters. Additionally, we provide extensive evaluations and analyses demonstrating the advantages of our proposed scaling framework over existing dense or token-choice approaches. As noted in our previous response, we are actively working on additional comparative experiments and will update the paper to further demonstrate our findings.
>
> >   Can the authors make the notations in Section 2.1 regarding the rectified flow clearer? Please explain p0, x0, xt, v_theta in more details.
>
> **Response:**  We are sorry for the confusion. Here, $x_0\sim p_0(x)$ represents the image sample and data distribution, respectively. We denote pure noise and its corresponding distribution as $x_1\sim p_1=\mathcal{N}(0, I)$. In rectified flow, the forward process is defined as a linear path between the data distribution $p_0$ and noise $p_1$, specifically $x_t=(1-t)x_0+tx_1$ [1-3]. This construction induces an ODE over the timestep $t\in[0,1]$, i.e., $dx_t=v_\theta(x_t, t)dt$. Here, $v_\theta(x_t, t)$ represents the velocity direction of the path, parameterized by the DiT backbone. Additionally, since $x_t=(1-t)x_0+tx_1$, it follows that $v_\theta(x_t, t)=\frac{dx_t}{dt}=x_1-x_0$, leading to the L-2 loss in equation 1.
>
> We have updated our paper to provide a more detailed explanation and corrected typos in Section 2.1.

---

> ### Author Response · Authors · 2024-11-21
>
> >   We know that DiT performs the diffusion in the latent space instead of the actual image pixel space. Does it undermine the motivation of this paper to allocate more computations to image regions of higher complexity? Please explain how working in latent space affects their approach to computational allocation.
>
> **Response:** The autoencoder used in DiT [4] and Latent Diffusion Models (LDMs) [5] learns a latent space that is perceptually equivalent to the image space while significantly reducing computational complexity. In other words, the autoencoder performs perceptual compression of the pixel space, retaining global image information in a low-dimensional latent space. As discussed in [5], the autoencoder is pretrained with a slight KL regularization (KL-reg) by a factor of approximately $10^{-6}$, which enables high-fidelity reconstructions on perceptible patterns. This setup allows us to efficiently allocate computation based on the compressed and preserved image information within the latent space. As illustrated in Figure 6, EC-DiT leverages this property to effectively allocate more computational resources to significant objects or areas of higher complexity.
>
> >   From Section 2.1, it seems that the authors use the Rectified Flow method on the DiT architecture with MoE. Can the authors explain why they chose to use Rectified Flow and how it integrates with their overall approach? This would help clarify the role of Rectified Flow in their method.
>
> **Response:** Rectified flow constructs a straightforward linear path between data and noise distributions, which gets rid of the mathematical complexity of SDE models and lends it conceptual simplicity [3]. Concurrent works [5-7] have also demonstrated rectified flow’s strong generation quality and scalability. In this work, we leverage rectified flow for text-to-image synthesis by parameterizing the velocity with the DiT backbone. Additional details on this are provided in our earlier response, complementary to Section 2.1.
>
> We would also like to emphasize that the proposed adaptive computation and scaling framework is generative-approach-agnostic. The context-aware routing technique can be seamlessly integrated into diffusion-based methods, such as DDPM. We will explore how our approach performs with these alternative generative paradigms for future work.
>
> **Reference:**
>
> [1] Patrick Esser, Sumith Kulal, Andreas Blattmann, Rahim Entezari, Jonas Muller, Harry Saini, Yam Levi, Dominik Lorenz, Axel Sauer, Frederic Boesel, Dustin Podell, Tim Dock horn, Zion English, Kyle Lacey, Alex Goodwin,Yannik Marek, and Robin Rombach. Scaling rectified flow transformers for high-resolution image synthesis, 2024.
>
> [2] Albergo, M. S. and Vanden-Eijnden, E. Building normaliz-ing flows with stochastic interpolants, 2022.
>
> [3] Xingchao Liu,Chengyue Gong, and QiangLiu. Flow Straight and Fast: Learning to Generate and Transfer Data with Rectified Flow. arXiv,2022.doi: 10.48550/arxiv.2209.03003.
>
> [4] Peebles, William S. and Saining Xie. “Scalable Diffusion Models with Transformers.” 2023 IEEE/CVF International Conference on Computer Vision (ICCV) (2022): 4172-4182.
>
> [5] Rombach, R., Blattmann, A., Lorenz, D., Esser, P., & Ommer, B. (2021). High-Resolution Image Synthesis with Latent Diffusion Models. 2022 IEEE/CVF Conference on Computer Vision and Pattern Recognition (CVPR), 10674-10685.
>
> [6] Liu, X., Zhang, X., Ma, J., Peng, J., & Liu, Q. (2023). InstaFlow: One Step is Enough for High-Quality Diffusion-Based Text-to-Image Generation. ArXiv, abs/2309.06380.
>
> [7] Ma, N., Goldstein, M., Albergo, M.S., Boffi, N.M., Vanden-Eijnden, E., and Xie, S. 2024. SiT: Exploring Flow and Diffusion-based Generative Models with Scalable Interpolant Transformers. arXiv.

---

> > ### Comment · Reviewer_gySw · 2024-11-23
> >
> > Thanks to the authors for their responses. Most of my comments are addressed.

---

> > > ### Author Response · Authors · 2024-11-24
> > >
> > > Thank you very much for providing insightful feedback and taking the time to review our rebuttal.

---

### Author Response · Authors · 2024-11-21

We thank all Reviewers for their constructive feedback and valuable suggestions.

In this work, we introduce a context-aware scaling approach for DiT with adaptive computation. The proposed EC-DiT leverages global image information to achieve effective heterogeneous compute allocation. This approach demonstrates superior training convergence, improved text-to-image alignment, and enhanced generation quality compared to dense and conventional MoE models. Furthermore, it enables efficient scaling up to 97B parameters. Extensive evaluations show that EC-DiT delivers promising generation performance while maintaining competitive inference speed and offering intuitive interpretability.

In the revised draft, we have incorporated updates based on the Reviewers’ feedback, as summarized below:

-   Introduction: Refined for more precise language and better clarity.
-   Section 2.1: Expanded with a more detailed explanation of the rectified flow and the training objective.
-  Section 2.3 and Remark: Updated with clearer descriptions to address potential ambiguities.
-   Section 3.2 and Table 2: Added further discussions and clarifications regarding the GenEval evaluation.
-   Section 3.5 and Table 1: Revised details to improve readability and clarity.
-   Appendix A: Added detailed future directions inspired by the reviews.
-   Appendix B: Included DSG results for token-choice baselines to provide a more comprehensive comparison.
-   Appendix D: Added the pseudocode for the activated size calcualtion.
-  Appendix E: Added additional generated samples covering a variety of topics and art styles to better illustrate EC-DiT’s capabilities.
- Appendix E: Added side-by-side comparison with SD3 and FLUX.
-   Minor Revisions: Corrected typos and adopted more accurate phrasing in several parts of the paper, including lines 210, 273, and 295.

---

### Meta-Review · Area_Chair_LDRD · 2024-12-23

**Metareview:**

In this paper, authors propose mixture of experts for diffusion models. Specifically, a new family of Mixture-of-Experts (MoE) models with adaptive expert-choice routing is proposed. By leveraging the computational heterogeneity of image generation, EC-DIT adaptively allocates compute to understand input texts and generate image patches, enabling heterogeneous computation aligned with text-image complexity. Experimental results show good qualitative and quantitative performance improvement.

**Additional Comments On Reviewer Discussion:**

In the rebuttal, reviewers raised several concerns about performance differnece, generalizability, comparison with other DiT-MoE methods, missing details about memory usage and activation params, etc. Authors sufficiently addressed all the concerns in the rebuttal. All reviewers are leaning towards acceptance. Even though the method simply combines DiT and MoE, I feel it is a good contribution to the community by showing the benefits of applying expert-choice routing in DiT. So, I vote for accepting the paper.

---

### Decision · Program_Chairs · 2025-01-22

Accept (Poster)